# LoRI: Reducing Cross-Task Interference in Multi-Task Low-Rank Adaptation

**Juzheng Zhang**[1],    **Jiacheng You**[2],    **Ashwinee Panda**[1],    **Tom Goldstein**[1]
[1]University of Maryland    [2]Tsinghua University

## Abstract

Low-Rank Adaptation (LoRA) has emerged as a popular parameter-efficient fine-tuning (PEFT) method for Large Language Models (LLMs), yet it still incurs notable overhead and suffers from parameter interference in multi-task scenarios. We propose **Lo**RA with **R**educed **I**nterference (**LoRI**), a simple yet effective approach that freezes the projection matrices $A$ as random projections and sparsifies the matrices $B$ using task-specific masks. This design substantially reduces the number of trainable parameters while maintaining strong task performance. Moreover, LoRI minimizes cross-task interference in adapter merging by leveraging the orthogonality between adapter subspaces, and supports continual learning by using sparsity to mitigate catastrophic forgetting. Extensive experiments across natural language understanding, mathematical reasoning, code generation, and safety alignment tasks demonstrate that LoRI outperforms full fine-tuning and existing PEFT methods, while using up to 95% fewer trainable parameters than LoRA. In multi-task experiments, LoRI enables effective adapter merging and continual learning with reduced cross-task interference. Code is available at: https://github.com/juzhengz/LoRI.

## 1  Introduction

Large language models (LLMs) (Brown et al., 2020; Touvron et al., 2023; Chowdhery et al., 2023) have transformed deep learning, showcasing remarkable capabilities across various domains. However, their deployment remains computationally demanding, particularly when fine-tuning is required to adapt to downstream tasks or align with human preferences. To mitigate the high resource costs, researchers have developed a range of parameter-efficient fine-tuning (PEFT) techniques. Among these techniques, LoRA (Hu et al., 2021) has gained widespread adoption due to its compelling balance of performance and efficiency. Nevertheless, LoRA still introduces notable memory overhead, particularly in large-scale models. Consequently, recent research has focused on further optimizing LoRA by reducing the number of trainable parameters without compromising performance (Kopiczko et al., 2023; Ding et al., 2023; Zhang et al., 2023b).

Recent studies (Yu et al., 2024; Panda et al., 2024) have shown that delta parameters – the differences between fine-tuned and pretrained model weights – exhibit significant redundancy. Furthermore, previous works (Zhang et al., 2023b; Zhu et al., 2024) have observed that freezing matrices $A$ in LoRA often achieves comparable performance to training them. Motivated by these findings, we propose **Lo**RA with **R**educed **I**nterference (**LoRI**). LoRI keeps matrices $A$ fixed as random projections, while training matrices $B$ using task-specific sparse masks. To retain the most critical elements of $B$, LoRI performs a calibration process to extract sparse masks by selecting the highest-magnitude elements across all layers and projections. As shown in Figure 1(a), LoRI maintains performance even with 90% sparsity in $B$ while keeping $A$ frozen. This demonstrates that adaptation does not require updating $A$, and that $B$ has considerable redundancy. By applying more constrained updates than LoRA, LoRI significantly reduces the number of trainable parameters while better preserving the pretrained model's knowledge during adaptation.

---

Correspondence to: juzheng@umd.edu.

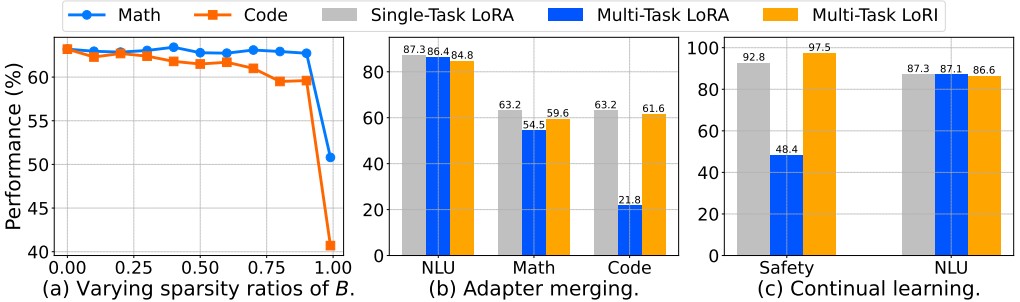

Figure 1: (a) Varying sparsity ratios in matrices $B$ while freezing $A$. Performance remains stable even at 90% sparsity in matrices $B$. (b) Merging three adapters via weighted averaging. LoRA suffers degradation due to parameter interference, while LoRI preserves task performance. (c) Continual learning from Safety to NLU. LoRA suffers from catastrophic forgetting, while LoRI retains safety alignment. Results for NLU are averaged over eight tasks. GSM8K accuracy (Math), HumanEval pass@10 (Code), and HEx-PHI refusal rate (Safety) are reported individually. Base model: Llama-3-8B, rank $r = 32$.

Multi-task learning is essential for enabling versatile models with multi-task capabilities, which is traditionally performed via joint training on a combination of task-specific datasets (Caruana, 1997; Sener & Koltun, 2018). However, training large models on this data mixture is prohibitively expensive in terms of time and compute. *Model merging* is a training-free alternative for building powerful models by combining existing ones (Ilharco et al., 2022; Yadav et al., 2023; Yu et al., 2024). This approach is well-suited for merging LoRA adapters, enabling multi-task capabilities within a single model during inference (Wang et al., 2024a; Prabhakar et al., 2024; Stoica et al., 2024). However, as shown in Figure 1(b), directly merging heterogeneous LoRAs often results in *parameter interference*, leading to degraded performance compared to single-task LoRAs. Additionally, many existing merging methods require trial-and-error to identify the optimal method for a specific combination of tasks. LoRI addresses these challenges by using fixed, randomly initialized projection $A$, which maps task-specific adapters into approximately orthogonal subspaces. This reduces interference when merging multiple adapters. In addition, LoRI enables adapter merging without manual selection of merging methods.

Beyond multi-tasking, safety-critical scenarios require that each newly introduced adapter enhances model capabilities while preserving the safety alignment of the pretrained base model (Qi et al., 2023). LoRI provides a lightweight *continual learning* approach for adapting models while preserving safety, where training is performed sequentially across tasks (Lopez-Paz & Ranzato, 2017; Wu et al., 2022; Ouyang et al., 2022). The strategy involves first fine-tuning an adapter on safety data to establish alignment, followed by separate adaptation to each downstream task. However, as illustrated in Figure 1(c), continual learning often leads to *catastrophic forgetting* (Li & Hoiem, 2017; Dong et al., 2023; Luo et al., 2023), wherein the adaptation to new tasks substantially compromises previously acquired knowledge. LoRI mitigates forgetting by leveraging the sparsity of projection $B$ through task-specific masks. This isolation of parameter updates across tasks facilitates continual learning with minimal interference, preserving both safety and task effectiveness.

To evaluate the effectiveness of LoRI, we conduct extensive experiments across a diverse suite of benchmarks spanning natural language understanding (NLU), mathematical reasoning, code generation, and safety alignment tasks. Using Llama-3-8B and Mistral-7B as base models, our results show that LoRI achieves performance comparable to – or better than – full fine-tuning (FFT), LoRA, and other PEFT methods, while using up to 95% fewer trainable parameters than LoRA. Notably, LoRI with 90% sparsity in $B$ surpasses LoRA by 17.3% on HumanEval with Llama-3. Beyond single-task adaptation, we evaluate LoRI in multi-task settings, including adapter merging and continual learning scenarios. Concatenated merging of LoRI adapters consistently outperforms LoRA adapters overall, closely matching the performance of single-task LoRA baseline. In continual learning, LoRI significantly outperforms LoRA in mitigating catastrophic forgetting of safety alignment, while maintaining strong performance on downstream tasks.

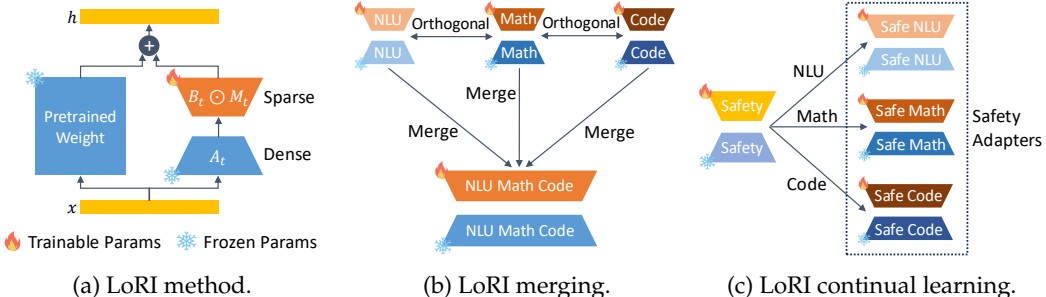

(a) LoRI method.  (b) LoRI merging.  (c) LoRI continual learning.

Figure 2: Overview of the proposed LoRI method. (a) LoRI freezes the projection matrices $A_t$ and sparsely updates $B_t$ using task-specific masks $M_t$. (b) LoRI enables adapter merging of multiple task-specific adapters with reduced parameter interference. (c) LoRI builds safety adapters by continual learning with reduced catastrophic forgetting.

## 2 Method

### 2.1 Freezing Low-Rank Projections with Sparse Masking

**Freezing Projection $A$.** LoRA (Hu et al., 2021) fine-tunes a weight update matrix as a product of two low-rank matrices to adapt LLMs to new tasks. Formally, for a specific task $t$, given a pretrained weight matrix $W_0 \in \mathbb{R}^{d_{in} \times d_{out}}$, the weight update $\Delta_t \in \mathbb{R}^{d_{in} \times d_{out}}$ is constrained to a low-rank decomposition:

$$h = xW_0 + x\Delta_t = xW_0 + xA_tB_t. \tag{1}$$

where $A_t \in \mathbb{R}^{d_{in} \times r}$, $B_t \in \mathbb{R}^{r \times d_{out}}$, and $r \ll \min\{d_{in}, d_{out}\}$. We denote $\Delta_t$ as the LoRA adapter for task $t$. In practice, LoRA adapters are typically applied to multiple projection matrices (e.g., $W_q$, $W_v$) within each transformer layer.

Typically, the low-rank projection matrices $A_t$ and the low-rank expansion matrices $B_t$ are updated via gradient descent. Matrices $A_t$ are usually initialized with Kaiming Uniform distribution (He et al., 2015), while matrices $B_t$ are initialized to zero, ensuring that $\Delta_t = 0$ at the start of training. However, in LoRI, we fix $A_t$ as random projections, meaning that the model only learns how to combine the fixed subspace via $B_t$. By freezing $A_t$, we eliminate the need to store their gradients and optimizer states, thereby reducing memory consumption. During inference, similar to LoRA, LoRI merges the low-rank updates by adding $A_tB_t$ to $W_0$, ensuring no additional inference latency compared to full fine-tuning.

**Sparse Masking for Projection $B$.** LoRI freezes matrices $A_t$ and selectively updates only the most relevant parameters in $B_t$ for each task, as illustrated in Figure 2(a). For task $t$, it first extracts sparse masks $M_t$ through a calibration process, then applies the masks to constrain training to a limited subset of parameters in $B_t$. During mask calibration, LoRI updates $B_t$ without masking using a calibration dataset $\mathcal{D}_t^C$, sampled from the adaptation dataset $\mathcal{D}_t$. After this phase, LoRI collects all $B_t$ matrices from the model across layers and projections. Then it computes a global threshold $\tau_t$, defined as the $s\%$ quantile of the absolute values of all elements from these matrices, where $s$ is the sparsity ratio. For each matrix $B_t$, the corresponding sparse mask $M_t$ is computed as:

$$M_t = \mathbb{I}\left(|B_t| \geq \tau_t\right), \quad \text{where} \quad \tau_t = \text{Quantile}_s\left(\bigcup |B_t|\right). \tag{2}$$

Here, $\mathbb{I}(\cdot)$ denotes the indicator function applied element-wise. This ensures that only the top-$(1-s)\%$ of parameters (by magnitude) across all layers and projections are retained. The masks can also be derived using gradient-based measures such as the Fisher information matrix (Guo et al., 2023; Iurada et al., 2025) or SNIP score (Lee et al., 2018). However, these methods capture local sensitivity at a specific training step, whereas magnitude reflects cumulative importance over the entire fine-tuning process.

It is well established that the importance of projection matrices varies significantly across different layers and projections (Zhang et al., 2023a;d; Kopiczko et al., 2023). Our masking strategy enables global comparison of parameters and facilitates effective allocation of the parameter budget determined by the sparsity ratio. Notably, the masks for each task $t$ are calibrated only once and can be reused as needed.

After mask calibration, LoRI resets $B_t$ to zero and trains on the adaptation dataset $\mathcal{D}_t$, with updates restricted to the masked parameters. The LoRI adapter is expressed as $\Delta_t = A_t(B_t \odot M_t)$. The algorithm of LoRI is detailed in Appendix B. In practice, the sparsity ratio $s$ can reach up to 90%, meaning that only a small fraction of parameters in matrices $B_t$ are updated, while the majority remain unchanged. This selective adaptation enables the model to focus on modifying the most critical parameters needed for specific tasks, while preserving the foundational knowledge encoded in the pretrained base model. In the limiting case of a single task and zero sparsity, our method reduces to LoRA-FA (Zhang et al., 2023b), which has been shown to perform competitively with standard LoRA.

## 2.2 Reducing Interference in Adapter Merging via Orthogonality

**Orthogonality of LoRI Adapters.**  A central challenge in adapter merging is *parameter interference*, where combining multiple adapters leads to degraded performance due to conflicting parameter updates. Given a set of trained LoRI adapters $\{\Delta_1, \Delta_2, \ldots, \Delta_T\}$, the goal is to construct a unified model that combines knowledge from all tasks with minimal interference, as illustrated in Figure 2(b). Formally, we define the excess loss due to parameter interference for a specific task $t$ as:

$$\mathcal{I}_t = \mathcal{L}_t(W_{\mathrm{merge}}) - \mathcal{L}_t(W_0 + \alpha_t \Delta_t), \tag{3}$$

where $W_{\mathrm{merge}}$ is the merged model, $W_0$ is the pretrained weight matrix, $\Delta_t$ is the LoRI adapter for task $t$, $\alpha_t \in \mathbb{R}$ is a scalar weight, and $\mathcal{L}_t$ is the loss function for task $t$. A high $\mathcal{I}_t$ indicates significant interference.

LoRI mitigates this interference by leveraging *approximate orthogonality*, achieved by freezing the projection matrices $A_t$ as independent random matrices. This design leads to the following property, whose proof is provided in Appendix C:

**Property 1.** *Let $A_s, A_t \in \mathbb{R}^{d_{in} \times r}$ be independent random matrices with i.i.d. entries drawn from a Kaiming Uniform distribution for distinct tasks $s \neq t$. Let their corresponding LoRI adapters be $\Delta_s = A_s(B_s \odot M_s)$ and $\Delta_t = A_t(B_t \odot M_t)$, where the trained matrices $(B_s \odot M_s)$ and $(B_t \odot M_t)$ have finite Frobenius norms. Under the condition that $r \ll d_{in}$, as the input dimension $d_{in} \to \infty$, the adapters are approximately orthogonal:*

$$\langle \Delta_s, \Delta_t \rangle_F \to 0 \quad \textit{in probability.} \tag{4}$$

We describe two merging methods: concatenated merging (weighted averaging) and linear merging (Task Arithmetic) (Ilharco et al., 2022), both of which exploit the approximate orthogonality of LoRIs.

**Concatenated Merging (Weighted Averaging).**  This method constructs the merged model by creating a weighted sum of individual task adapters. This is achieved by concatenating the weighted $A$ and masked $B$ matrices:

$$A' = [\alpha_1 A_1 \; \alpha_2 A_2 \; \ldots \; \alpha_T A_T], \quad B' = \left[(B_1 \odot M_1)^\top, \ldots, (B_T \odot M_T)^\top\right]^\top, \tag{5}$$

where $\alpha_t \in \mathbb{R}$ are scalar weights (e.g., uniform or task-prioritized). The final merged model is then formed by adding their product to the base model weights:

$$W_{\mathrm{merge}} = W_0 + A'B' = W_0 + \sum_{t=1}^{T} \alpha_t A_t(B_t \odot M_t) = W_0 + \sum_{t=1}^{T} \alpha_t \Delta_t. \tag{6}$$

By summing approximately orthogonal adapters, we ensure that the updates for each task occupy largely disjoint subspaces, thereby reducing interference (Ilharco et al., 2022; Ortiz-Jimenez et al., 2023; Xiong et al., 2024).

The reduction in interference can be explained by a theoretical sketch based on two key assumptions. The first is the local linearity of the loss landscape (Li et al., 2018), which allows for a first-order Taylor approximation. The second is the gradient alignment assumption, formally expressed as $\nabla \mathcal{L}_t(W_0 + \alpha_t \Delta_t) \propto \Delta_t$. This posits that at a task's solution, the direction of steepest descent is primarily aligned with the adapter updates already made for that task. Under these assumptions, the excess loss $\mathcal{I}_t$ is approximately the inner product of the gradient and the updates from the other tasks:

$$\mathcal{I}_t \approx \left\langle \nabla \mathcal{L}_t(W_0 + \alpha_t \Delta_t), \sum_{s \neq t} \alpha_s \Delta_s \right\rangle_F \propto \sum_{s \neq t} \alpha_k \langle \Delta_t, \Delta_s \rangle_F. \tag{7}$$

Since Property 1 establishes that $\langle \Delta_t, \Delta_s \rangle_F \to 0$ for $s \neq t$, the total interference loss becomes negligible: $\mathcal{I}_t \approx 0$. This heuristic argument provides strong intuition for why concatenated merging is effective, which is then validated by our empirical results.

**Linear Merging (Task Arithmetic).**  Alternatively, the merged model can be formed by summing the $A_t$ and masked $B_t$ matrices independently before multiplication:

$$W_{\text{merge}} = W_0 + \left( \sum_{t=1}^{T} \alpha_t A_t \right) \left( \sum_{t=1}^{T} \alpha_t (B_t \odot M_t) \right) = W_0 + \sum_{s=1}^{T} \sum_{t=1}^{T} \alpha_s \alpha_t A_s (B_t \odot M_t). \tag{8}$$

While concatenated merging directly sums approximately orthogonal adapters, this linear merging approach introduces problematic cross-terms $\alpha_s \alpha_t A_s (B_t \odot M_t)$ for $s \neq t$. These terms cause interference because components like $\{A_s(B_t \odot M_t)\}_{t=1}^{T}$ for a fixed $s$ are generally not mutually orthogonal. As a result, concatenated merging offers a cleaner and empirically more effective strategy for combining LoRI adapters.

## 2.3 Reducing Interference in Continual Learning via Sparsity

**Safety-Preserving Adapters.**  For safety-critical applications, ensuring that new task adaptations do not compromise established safety behaviors is crucial. Therefore, each newly introduced adapter must preserve the base model's safety alignment. A straightforward approach to achieve this is to merge a safety LoRI adapter into the deployed model during every inference. However, as we will show in Section 3.4, this method may be insufficient for scenarios that demand strong safety guarantees. In such cases, as illustrated in Figure 2(c), a more reliable solution is to adopt a two-phase continual learning process for each LoRI adapter to reinforce safety:

1. **Safety Alignment Phase:** Train a LoRI adapter on a curated safety dataset $\mathcal{D}_{\text{safety}}$, yielding $\Delta_{\text{safety}} = A(B_{\text{safety}} \odot M_{\text{safety}})$.

2. **Task Adaptation Phase:** Fine-tune $\Delta_{\text{safety}}$ on each task adaptation dataset $\mathcal{D}_t, t = 1, 2, \ldots, T$, reusing the calibrated task-specific masks $M_t$, resulting in safety-preserving adapters $\Delta_t = A(B_t \odot M_t)$.

This method does not require recalibrating masks for each task or performing multiple rounds of continual learning. Notably, we do not enforce non-overlapping masks $M_t \cap M_{\text{safety}} = \varnothing$. Enforcing such a constraint would require recalibrating masks after the safety alignment phase due to the reduced parameter space, and could potentially degrade performance on downstream tasks. The expected overlap between sparse masks with 90% sparsity is theoretically 1%. Empirically, we find that this expectation holds: the average overlap between task-specific masks is indeed $\sim 1\%$, without explicitly enforcing non-overlap. This slight overlap allows important parameters to be shared across tasks, potentially enabling positive knowledge transfer.

**Catastrophic Forgetting.**  Continual learning models are vulnerable to *catastrophic forgetting* (Li & Hoiem, 2017; Dong et al., 2023; Luo et al., 2023), where updates for new tasks can overwrite and degrade previously learned knowledge. Despite the slight overlap between

task-specific masks, the *sparsity* in $B_t$ induced by $M_t$ enables LoRI to facilitate isolated parameter updates for safety alignment and task adaptation. As a result, LoRI minimizes cross-task interference and mitigates catastrophic forgetting in safety alignment.

# 3 Experiments

## 3.1 Experimental Setup

**Datasets.** We conduct a series of experiments to evaluate LoRI's effectiveness on single-task and multi-task settings, including adapter merging and continual learning. We focus on four capabilities: (i) Natural Language Understanding (**NLU**): LoRI is trained on the aggregation of eight NLU datasets (Hu et al., 2023), including BoolQ (Clark et al., 2019), PIQA (Bisk et al., 2020), SocialIQA (Sap et al., 2019), ARC-Challenge (Clark et al., 2018), ARC-Easy (Clark et al., 2018), OpenBookQA (Mihaylov et al., 2018), HellaSwag (Zellers et al., 2019), and Winogrande (Sakaguchi et al., 2021). We evaluate accuracy on the individual test split for each dataset. (ii) Mathematical Reasoning (**Math**): LoRI is trained on the GSM8K (Cobbe et al., 2021) training split and evaluated on the GSM8K test split. (iii) Code Generation (**Code**): LoRI is trained on CodeAlpaca (Chaudhary, 2023) and evaluated using pass@1, pass@5, and pass@10 on HumanEval (Chen et al., 2021). (iv) Safety Alignment (**Safety**): LoRI is trained on Saferpaca (Bianchi et al., 2023), which extends Alpaca-Cleaned (Taori et al., 2023) with 2,000 safety instructions. Safety performance is assessed by measuring the refusal rate on harmful queries from HEx-PHI (Qi et al., 2023).

**Baselines.** In single-task experiments, we compare LoRI with full fine-tuning (FFT), LoRA (Hu et al., 2021), and DoRA (Liu et al., 2024). Results for additional PEFT baselines, including VeRA (Kopiczko et al., 2023), IA3 (Liu et al., 2022), LoRA-FA (Zhang et al., 2023b), AdaLoRA (Zhang et al., 2023d), rsLoRA (Kalajdzievski, 2023), PiSSA (Meng et al., 2024), and LoRA+ (Hayou et al., 2024), are available in Appendix E.1. In merging experiments, we compare LoRI merging with several LoRA merging methods, including concatenated merging, linear merging (Ilharco et al., 2022), magnitude pruning, TIES-Merging (Yadav et al., 2023), and DARE (Yu et al., 2024). Magnitude pruning, TIES, and DARE are pruning-based approaches that apply sparsification to the $A$ and $B$ matrices before merging, based on a specified density. Magnitude pruning removes low-magnitude parameters; TIES-Merging further merges weights with consistent signs; and DARE performs random pruning followed by rescaling. For fair comparison, all baseline results are reproduced using a consistent experimental setup.

**Implementation Details.** We use Llama-3-8B (Grattafiori et al., 2024) and Mistral-7B (Jiang et al., 2023) as base models. We conduct all experiments on 8 NVIDIA A5000 GPUs. To explore the impact of sparsity, we provide two variants of LoRI: **LoRI-D**, which uses dense $B$ matrices, and **LoRI-S**, which applies 90% sparsity to $B$. Sparsity is implemented by masking the gradients of $B$ during backpropagation. For optimal performance, we use the entire adaptation dataset as the calibration dataset for each task. Ablation results for calibration are presented in Section 3.5. For consistency, we use the same hyperparameters for PEFT baselines as for LoRI-D. For all adapter merging experiments, uniform weights $\alpha_t$ are employed across all adapters. The weights $\alpha_t$ are treated as hyperparameters, and their ablation study is detailed in Section 3.5. Detailed hyperparameter settings are provided in Appendix D.

## 3.2 Single-Task Performance

Table 1 presents single-task performance on eight NLU benchmarks, while Table 2 reports single-task performance on the math, code, and safety benchmarks. Results for additional PEFT baselines are available in Appendix E.1. The rank for our experiments is set to $r = 32$. We observed stable performance across different ranks, with additional results for $r = 64$ provided in Appendix E.2.

Table 1: Performance comparison of different adaptation methods on eight NLU benchmarks using Llama-3 and Mistral with $r = 32$. **Bold** indicates the best-performing method, and underline indicates the second-best.

| Method | # Params (%) | BoolQ | PIQA | SIQA | ARC-c | ARC-e | OBQA | HellaS | WinoG | Avg. |
|---|---|---|---|---|---|---|---|---|---|---|
| *Llama-3-8B* | | | | | | | | | | |
| FFT | 8.03G (100%) | 73.8 | 86.8 | 77.6 | 76.7 | 87.6 | 84.1 | 93.2 | 85.1 | 83.1 |
| LoRA | 84M (1.03%) | 76.3 | **89.8** | 82.7 | 83.4 | 91.7 | 88.4 | 95.8 | **88.7** | 87.1 |
| DoRA | 85M (1.05%) | 75.9 | **89.8** | 82.7 | 83.5 | 93.2 | 87.9 | 95.3 | 88.2 | 87.1 |
| LoRI-D | 44M (0.54%) | **76.4** | 89.0 | 82.7 | **84.2** | **93.6** | **88.5** | **95.9** | 87.9 | **87.3** |
| LoRI-S | **4.4M (0.05%)** | 75.2 | 89.2 | **82.8** | 83.8 | 92.6 | 88.4 | 95.2 | 87.5 | 86.8 |
| *Mistral-7B* | | | | | | | | | | |
| FFT | 7.24G (100%) | 74.1 | 84.6 | 78.0 | 79.3 | 90.5 | 88.4 | 94.4 | 83.5 | 84.1 |
| LoRA | 84M (1.15%) | 75.2 | 90.1 | 82.9 | 82.9 | 92.0 | 88.7 | 95.1 | **88.1** | 86.9 |
| DoRA | 85M (1.16%) | 75.8 | 90.4 | 82.9 | 83.3 | **92.6** | 90.6 | **96.3** | 87.9 | **87.5** |
| LoRI-D | 44M (0.60%) | **75.9** | **90.6** | **83.0** | **83.6** | 91.9 | 88.4 | 95.9 | 87.4 | 87.1 |
| LoRI-S | **4.4M (0.06%)** | 74.0 | 90.1 | 82.6 | 82.6 | 91.5 | **90.8** | 95.5 | 87.5 | 86.8 |

Table 2: Performance comparison of different adaptation methods on GSM8K (math), HumanEval (code), and HEx-PHI (safety) benchmarks using Llama-3 and Mistral with $r = 32$. **Bold** indicates the best-performing method, and underline indicates the second-best.

| Method | # Params (%) | GSM8K | HumanEval | | | HEx-PHI |
|---|---|---|---|---|---|---|
| | | | Pass@1 | Pass@5 | Pass@10 | |
| *Llama-3-8B* | | | | | | |
| FFT | 8.03G (100%) | 58.8 | 30.5 | 39.3 | 41.7 | **94.8** |
| LoRA | 84M (1.03%) | 64.4 | 34.7 | 46.4 | 50.8 | 91.6 |
| DoRA | 85M (1.05%) | **65.4** | 33.1 | 44.0 | 48.6 | 93.6 |
| LoRI-D | 44M (0.54%) | 63.2 | **43.2** | **57.6** | **63.2** | 92.8 |
| LoRI-S | **4.4M (0.05%)** | 62.7 | 41.3 | 54.4 | 59.6 | 93.8 |
| *Mistral-7B* | | | | | | |
| FFT | 7.24G (100%) | 55.5 | 29.1 | 38.5 | 40.4 | 94.1 |
| LoRA | 84M (1.15%) | 57.8 | **33.8** | 42.4 | 45.3 | 91.9 |
| DoRA | 85M (1.16%) | 57.5 | 33.7 | 42.6 | 46.8 | 95.3 |
| LoRI-D | 44M (0.60%) | **58.0** | **33.8** | 42.0 | 45.1 | 94.7 |
| LoRI-S | **4.4M (0.06%)** | 57.1 | 33.7 | **43.6** | **48.1** | **95.9** |

While full fine-tuning (FFT) updates all model parameters, LoRA and DoRA reduce the number of trainable parameters to approximately 1%. LoRI-D further reduces this to about 0.5% by freezing matrices $A$, and LoRI-S pushes this reduction to 0.05% by applying 90% sparsity to matrices $B$, achieving a 95% reduction in trainable parameters compared to LoRA. Despite tuning fewer parameters, LoRI-D and LoRI-S achieve performance comparable to – and even better than – LoRA and DoRA on NLU, math, code, and safety tasks. LoRI-D generally outperforms LoRI-S slightly, due to the extremely limited parameter budget in LoRI-S. Remarkably, LoRI-D and LoRI-S consistently outperform FFT, LoRA, and DoRA on code generation tasks. On HumanEval with Llama-3, LoRI-D achieves a pass@10 score of 63.2%, outperforming LoRA by 24.4%. LoRI-S achieves 59.6% pass@10, exceeding LoRA by 17.3%.

The strong performance of LoRI-D suggests that effective adaptation can be achieved without updating $A$, while the strong performance of LoRI-S indicates that $B$ contains substantial parameter redundancy. LoRI's performance gains are attributed to the principled use of sparsity, which serves as a strong regularizer during adaptation. Additionally, LoRI preserves latent task-specific knowledge embedded in the pretrained model. This supports the view that supervised fine-tuning (SFT) primarily unlocks capabilities already present in pretrained models, rather than introducing new ones, which is consistent with findings from Liu et al. (2024); Yu et al. (2024).

Table 3: Comparison of merging methods for combining four adapters, evaluated on their respective benchmarks. The best-performing single-task adapter, LoRI-D, is used as the single-task baseline. Results for NLU are averaged over eight tasks. Base model: Llama-3-8B, rank $r = 32$. **Bold** indicates the best-performing method, and underline indicates the second-best.

| Merging | Adaptation | NLU | GSM8K | HumanEval | | | HEx-PHI |
| | | | | Pass@1 | Pass@5 | Pass@10 | |
|---|---|---|---|---|---|---|---|
| Single-Task | LoRI-D | 87.3 | 63.2 | 43.2 | 57.6 | 63.2 | 92.8 |
| Concat | LoRA | **85.0** | **57.8** | 13.0 | 20.0 | 22.3 | 84.4 |
| Linear | LoRA | 84.8 | 54.1 | 14.2 | 20.8 | 23.3 | 79.4 |
| Magnitude | LoRA | 81.9 | 50.3 | 24.1 | 36.7 | 42.4 | 74.4 |
| TIES | LoRA | 72.6 | 24.0 | 32.5 | 46.3 | 51.7 | 77.8 |
| DARE | LoRA | 79.1 | 48.9 | 34.1 | 48.7 | 53.5 | 74.1 |
| Concat | LoRI-D | 83.2 | 55.8 | 40.5 | **56.9** | **62.2** | **86.6** |
| Linear | LoRI-D | 82.5 | 53.8 | **40.9** | 54.9 | 60.3 | 85.9 |
| Concat | LoRI-S | 81.2 | 45.2 | 34.3 | 48.7 | 54.0 | 84.7 |
| Linear | LoRI-S | 79.1 | 41.3 | 23.2 | 36.6 | 42.3 | 78.8 |

## 3.3 Adapter Merging

We consider four heterogeneous tasks for LoRA and LoRI merging: NLU, math, code, and safety. This setting is generally more challenging than merging homogeneous adapters, such as merging multiple NLU adapters. Table 3 presents results for merging LoRAs and LoRIs on these four tasks. For LoRI, we apply concatenated and linear merging to the LoRI-D and LoRI-S variants. Pruning-based methods such as magnitude pruning, TIES, and DARE are not applied to LoRI, since these methods will prune the $A$ matrices as LoRI already sparsifies $B$, resulting in an inconsistent pruning scheme across $A$ and $B$. Additional results, including experiments on merging three adapters and evaluations of pruning-based methods on LoRI, are provided in Appendix E.4 and E.5.

As shown in Table 3, directly merging LoRAs results in substantial performance degradation, particularly for code generation and safety alignment. Although pruning-based methods (e.g., DARE, TIES) improve code performance, they often compromise accuracy on other tasks. In contrast, LoRI achieves consistently strong performance across all tasks.

Concatenated merging with LoRI-D achieves the best overall performance, closely matching the single-task baseline, which indicates minimal interference between LoRI adapters. For instance, it achieves 62.2% pass@10 on HumanEval and an 86.6% refusal rate on HEx-PHI. Despite using only 5% of the parameters of LoRA, LoRI-S retains competitive performance. Notably, on code and safety tasks, concatenated merging with LoRI-S outperforms all LoRA merging methods.

Linear merging with LoRI also performs competitively, though it lags slightly behind concatenated merging due to cross-term interactions that introduce some interference. LoRI eliminates the need for manual selection of merging methods: simple concatenated merging yields strong results. The choice between LoRI-D and LoRI-S can then be guided by the desired trade-off between performance and parameter efficiency. We also note an important trade-off between code generation performance and other domains during adapter merging, a phenomenon further explored in Section 3.5.

## 3.4 Continual Learning

While merging adapters enables multi-task capabilities, it falls short of providing robust safety alignment in scenarios that demand strong safety guarantees. As shown in Table 3, the highest refusal rate on HEx-PHI achieved through LoRA or LoRI merging is 86.6%. To address this limitation, we adopt a two-phase training process: first, a safety adapter is trained on the safety alignment dataset Saferpaca; then, it is individually adapted to each downstream task, including NLU, math, and code.

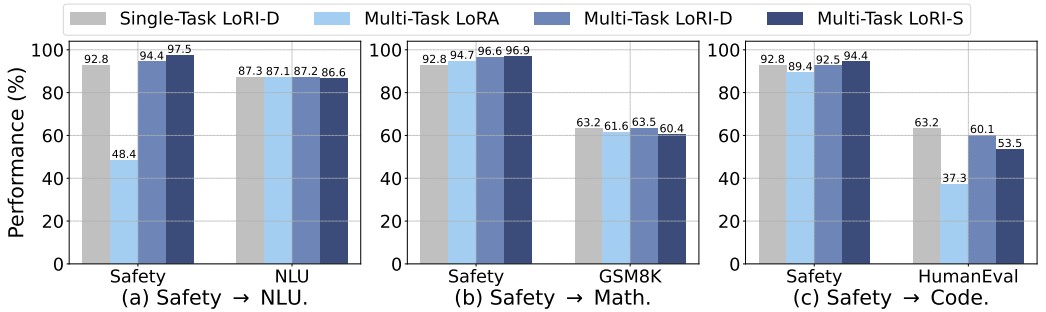

Figure 3: Continual learning results from safety to NLU, math, and code domains. Results for NLU are averaged over eight tasks. GSM8K accuracy, HumanEval pass@10, and HEx-PHI refusal rate are reported individually. Base model: Llama-3-8B, rank $r = 32$.

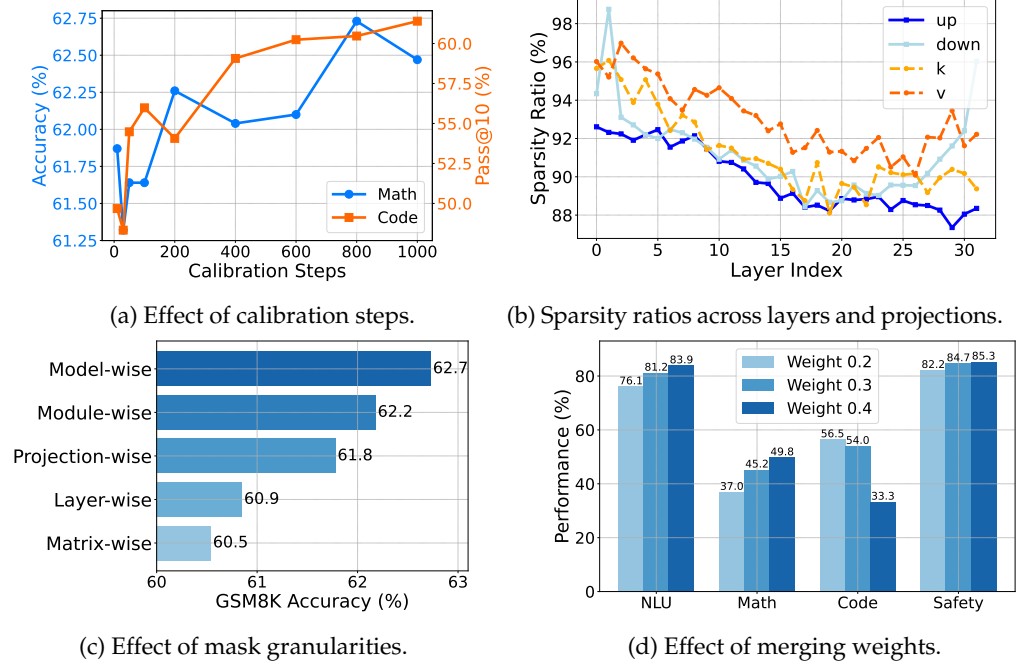

Figure 4: Ablation studies across different settings. Base model: Llama-3-8B, rank $r = 32$. Additional ablation studies are provided in Appendix F.

Figure 3 presents results from these continual learning experiments. LoRA exhibits severe catastrophic forgetting on safety alignment – particularly in the safety → NLU experiment – likely due to the large size of the NLU training split (~170k examples). Among all methods, LoRI-S achieves the best preservation of safety alignment, even outperforming single-task LoRI-D. This is due to its 90% sparsity in the $B$ matrices, which enables isolated parameter updates between the initial safety alignment and subsequent task adaptations. LoRI-D also shows some resistance to forgetting, benefiting from frozen $A$ matrices. For task adaptation, LoRI-D generally outperforms LoRI-S, as the latter's aggressive sparsity limits its adaptation capacity. Overall, LoRI offers a lightweight and effective approach to building safety adapters that preserve alignment while supporting adaptation to downstream tasks.

### 3.5 Ablation Studies

**Calibration Steps.** Calibration steps refer to the number of update steps used to generate sparse masks for each task. Figure 4(a) shows how performance of LoRI-S changes with

different numbers of calibration steps on math and code tasks. We observe that performance generally improves as the number of calibration steps increases. Since the masks only need to be calibrated once per task and can be reused, we use the entire adaptation dataset as the calibration dataset to achieve the best performance.

**Sparsity Ratio.** We use model-wise masks in our experiments that retain the highest-magnitude parameters across all layers and projections. Figure 4(b) presents the sparsity ratios of different projection types (e.g., up, down, key, value) across layers under a 90% sparsity on GSM8K. We observe that feedforward (FFN) projections tend to retain more parameters (i.e., lower sparsity) than self-attention projections, indicating they are more critical for adaptation. Additionally, the top layers are less sparse than lower layers, suggesting that the top layers play a more important role in adaptation.

**Mask Granularity.** We compare five levels of mask granularity under 90% sparsity on GSM8K, as shown in Figure 4(c). We compare module-wise, projection-wise, layer-wise, and matrix-wise masking against our model-wise masking, where parameters are selected within progressively smaller scopes. We find that coarse-grained masking (e.g., model-wise) yields the best performance, while fine-grained masking (e.g., matrix-wise) results in degradation. This suggests that global magnitude-based selection enables better parameter allocation, as the importance of projection matrices varies across the model.

**Merging Weights.** We adopt uniform weights across all adapters for adapter merging, rather than task-specific weights, as we do not wish to prioritize any individual task. Figure 4(d) shows the effect of different merging weights (0.2, 0.3, 0.4) for concatenated merging with LoRI-S. We observe that LoRI is moderately sensitive to merging weights, with a noticeable trade-off between performance on code tasks and other domains. We adopt 0.3 for all adapters in LoRI-S merging, as it offers a balanced performance across domains.

## 4 Conclusion

In this work, we introduced LoRI, a simple yet effective approach to parameter-efficient fine-tuning (PEFT) that substantially reduces trainable parameters while minimizing cross-task interference. By freezing the projection matrices $A$ as random projections and sparsifying $B$ using task-specific masks, LoRI achieves strong single-task performance across diverse domains – including natural language understanding, mathematical reasoning, code generation, and safety alignment – while reducing trainable parameters by up to 95% compared to LoRA. Furthermore, LoRI enables training-free adapter merging with minimal performance degradation, and supports continual learning with significantly reduced catastrophic forgetting. It also provides a lightweight approach to building safety adapters that preserve the safety alignment of the base model.

**Future Work.** We identify several promising avenues for extending this work. While LoRI currently leverages unstructured magnitude-based sparsity, future research can explore structured sparsity patterns – such as block sparsity, head pruning, or group-wise masking – which may offer better hardware compatibility. Additionally, although this study focuses on LLMs, the core design of LoRI is modality-agnostic. Extending LoRI to diffusion and vision-language models for multi-modal generation is a promising direction.

## Acknowledgements

This material is based upon work partially supported by the NSF Grant No. 2229885 (NSF Institute for Trustworthy AI in Law and Society, TRAILS). Any opinions, findings and conclusions or recommendations expressed in this material are those of the author(s) and do not necessarily reflect the views of the National Science Foundation.

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

# A   Related Works

**Parameter-Efficient Fine-Tuning.**   Parameter-efficient fine-tuning (PEFT) methods for LLMs (Houlsby et al., 2019; Pfeiffer et al., 2020; Li & Liang, 2021; Lester et al., 2021; Liu et al., 2021; Hu et al., 2021) have received increasing attention in recent years. Among them, LoRA (Hu et al., 2021), which introduces trainable low-rank matrices, has become one of the most widely adopted PEFT methods due to its strong performance and efficiency. LoRI is motivated by reducing parameter redundancy in LoRA through an asymmetric design: we freeze the projection matrices $A$ and enforce sparsity on the matrices $B$. Our work is closely related to several lines of research. In terms of *parameter efficiency*, our goal is shared by methods such as IA3 (Liu et al., 2022), VeRA (Kopiczko et al., 2023), and FourierFT (Gao et al., 2024). More specifically, our approach builds on the concept of *asymmetric LoRA variants*, which has been explored in works like LoRA-FA (Zhang et al., 2023b), AsymmetryLoRA (Zhu et al., 2024), and HydraLoRA (Tian et al., 2024). However, LoRI is distinct from these works by uniquely combining frozen $A$ with sparsely updated $B$. This targeted, asymmetric pruning of only the $B$ matrices also differentiates our method from general *LoRA pruning* techniques like Loraprune (Zhang et al., 2023c), LoRA-drop (Zhou et al., 2024), and SoRA (Ding et al., 2023), as well as SVD-based approaches such as AdaLoRA (Zhang et al., 2023d) and PiSSA (Meng et al., 2024).

**Model Merging.**   Achieving multi-task capabilities typically involves training on a mixture of diverse task datasets (Caruana, 1997; Sener & Koltun, 2018), which is often prohibitively expensive in time and compute. As an alternative, model merging has gained attention for combining multiple task-specific models into a single model (Matena & Raffel, 2022; Ilharco et al., 2022; Yadav et al., 2023; Yu et al., 2024). Fisher Merging (Matena & Raffel, 2022) uses weights from the Fisher information matrix to combine parameters, while Task Arithmetic (Ilharco et al., 2022) employs predefined scaling factors. TIES-Merging (Yadav et al., 2023) prunes low-magnitude parameters and merges those with consistent signs, and DARE (Yu et al., 2024) applies random pruning with rescaling. However, identifying the optimal merging method often requires trial and error. More recently, there has been growing interest in merging task-specific LoRA adapters (Chronopoulou et al., 2023; Huang et al., 2023; Wu et al., 2024; Wang et al., 2024a; Prabhakar et al., 2024; Stoica et al., 2024), often utilizing Mixture-of-Experts (MoE) architectures. Nonetheless, these methods typically require additional training to coordinate the adapters effectively. In contrast, LoRI eliminates the need for manual selection of merging methods or additional training. By ensuring approximate orthogonality between adapters, LoRI minimizes interference and preserves task-specific performance.

**Catastrophic Forgetting.**   Catastrophic forgetting is a fundamental challenge in continual learning (McCloskey & Cohen, 1989; Ramasesh et al., 2021; Liang et al., 2023; Wang et al., 2024b), where neural networks struggle to retain previously learned knowledge when adapting to new tasks. Wu et al. (2022) analyzed this phenomenon using layer-wise and task-wise probing to assess knowledge retention across tasks. Several studies (Dong et al., 2023; Luo et al., 2023) have empirically examined catastrophic forgetting in the continual fine-tuning of LLMs. To mitigate catastrophic forgetting, various approaches have been proposed. Rehearsal-based methods (Rolnick et al., 2019; Shin et al., 2017) store or generate past data to reinforce prior knowledge during training. Parameter isolation methods (Rusu et al., 2016; Mallya & Lazebnik, 2018; Konishi et al., 2023; Panda et al., 2024) allocate separate subnetworks or sparsely mask parameters for different tasks to prevent interference. Additionally, O-LoRA (Wang et al., 2023) learns tasks in distinct low-rank subspaces while ensuring orthogonality between them. LoRI falls under the category of parameter isolation methods, leveraging sparse task-specific masks to mitigate catastrophic forgetting during continual learning.

# B   Algorithm of LoRI

The full procedure of LoRI is summarized in Algorithm 1.

---

Algorithm 1: LoRA with Reduced Interference (LoRI)

---

**Require:** Task $t$, mask calibration dataset $\mathcal{D}_t^C$, adaptation dataset $\mathcal{D}_t$, sparsity ratio $s$, model $f$, loss function $\mathcal{L}_t$, learning rate $\eta_t$
 1: **for** each layer $l = 1, \ldots, L$ **do**
 2:     **for** each projection $m = 1, \ldots, M$ **do**
 3:         **Initialize:** $A_t^{(l,m)} \in \mathbb{R}^{d_{\text{in}} \times r} \leftarrow \mathcal{U}(-\sqrt{\frac{3}{d_{\text{in}}}}, \sqrt{\frac{3}{d_{\text{in}}}})$, $B_t^{(l,m)} \in \mathbb{R}^{r \times d_{\text{out}}} \leftarrow 0$
 4:     **end for**
 5: **end for**
 6: **for** each batch $(x, y)$ sampled from $\mathcal{D}_t^C$ **do**                         ▷ Calibration steps
 7:     **for** each $(l, m)$ **do**
 8:         $B_t^{(l,m)} \leftarrow B_t^{(l,m)} - \eta_t \cdot \nabla_{B_t^{(l,m)}} \mathcal{L}_t(f(x, y; B_t^{(l,m)}))$
 9:     **end for**
10: **end for**
11: $\tau_t \leftarrow \text{Quantile}_s \left( \bigcup_{l,m} |B_t^{(l,m)}| \right)$                 ▷ Compute global threshold $\tau_t$
12: **for** each $(l, m)$ **do**
13:     $M_t^{(l,m)} \leftarrow \mathbb{I} \left( |B_t^{(l,m)}| \geq \tau_t \right)$         ▷ Generate mask for top-$(1-s)\%$ entries
14:     $B_t^{(l,m)} \leftarrow 0$                        ▷ Reset to zero before adaptation
15: **end for**
16: **for** each batch $(x, y)$ sampled from $\mathcal{D}_t$ **do**                    ▷ Adaptation steps
17:     **for** each $(l, m)$ **do**
18:         $B_t^{(l,m)} \leftarrow B_t^{(l,m)} - \eta_t \cdot \left( \nabla_{B_t^{(l,m)}} \mathcal{L}_t(f(x, y; B_t^{(l,m)})) \odot M_t^{(l,m)} \right)$
19:     **end for**
20: **end for**

---

# C   Proof of Property 1

*Proof.* Our goal is to show that the Frobenius inner product $\langle \Delta_s, \Delta_t \rangle_F$ converges to zero in probability. Let $\tilde{B}_s = B_s \odot M_s$ and $\tilde{B}_t = B_t \odot M_t$. The inner product is given by:

$$\langle \Delta_s, \Delta_t \rangle_F = \text{Tr}(\Delta_s^\top \Delta_t) = \text{Tr}(\tilde{B}_s^\top A_s^\top A_t \tilde{B}_t). \tag{9}$$

We will prove this by showing that the random matrix $X = A_s^\top A_t$ converges to the zero matrix in probability as $d_{\text{in}} \to \infty$.

Let $a_s^k, a_t^l \in \mathbb{R}^{d_{\text{in}}}$ be the $k$-th and $l$-th columns of $A_s$ and $A_t$, respectively. The entries of these vectors are i.i.d. from a Kaiming Uniform distribution $U[-a, a]$ where $a = \sqrt{3/d_{\text{in}}}$. This implies a mean of 0 and variance of $\sigma^2 = a^2/3 = 1/d_{\text{in}}$. An entry of $X$ is the inner product $X_{kl} = (a_s^k)^\top a_t^l = \sum_{i=1}^{d_{\text{in}}} (A_s)_{ik}(A_t)_{il}$.

Let $Z_i = (A_s)_{ik}(A_t)_{il}$. The terms $Z_i$ are i.i.d. with $\mathbb{E}[Z_i] = \mathbb{E}[(A_s)_{ik}]\mathbb{E}[(A_t)_{il}] = 0$. Each term is bounded: $|Z_i| \leq a^2 = 3/d_{\text{in}}$. We apply Hoeffding's inequality to the sum $\sum_{i=1}^{d_{\text{in}}} Z_i$, where each term lies in $[-3/d_{\text{in}}, 3/d_{\text{in}}]$:

$$\mathbb{P}(|X_{kl}| \geq t) = \mathbb{P}\left( \left| \sum_{i=1}^{d_{\text{in}}} Z_i \right| \geq t \right) \leq 2 \exp\left( \frac{-2t^2}{\sum_{i=1}^{d_{\text{in}}}(6/d_{\text{in}})^2} \right) = 2 \exp\left( \frac{-t^2 d_{\text{in}}}{18} \right). \tag{10}$$

We now bound the probability that any of the $r^2$ entries of $X$ exceeds a threshold $t$ using the union bound:

$$\mathbb{P}(\max_{k,l} |X_{kl}| \geq t) = \mathbb{P}\left( \bigcup_{k,l=1}^{r} \{|X_{kl}| \geq t\} \right) \leq \sum_{k,l=1}^{r} \mathbb{P}(|X_{kl}| \geq t) \leq 2r^2 \exp\left( \frac{-t^2 d_{\text{in}}}{18} \right). \tag{11}$$

Table 4: Hyperparameter settings for LoRI on NLU datasets.

| Method | LoRI-D | LoRI-S | LoRI-D | LoRI-S | LoRI-D | LoRI-S | LoRI-D | LoRI-S |
|---|---|---|---|---|---|---|---|---|
| Base Model | Llama-3 | Llama-3 | Llama-3 | Llama-3 | Mistral | Mistral | Mistral | Mistral |
| Rank $r$ | 32 | 32 | 64 | 64 | 32 | 32 | 64 | 64 |
| $\alpha$ | 64 | 64 | 128 | 128 | 64 | 64 | 128 | 128 |
| Sparsity Ratio | 0 | 0.9 | 0 | 0.9 | 0 | 0.9 | 0 | 0.9 |
| Learning Rate | 5e-5 | 5e-4 | 5e-5 | 1e-4 | 1e-5 | 1e-4 | 1e-5 | 1e-4 |
| Dropout | | | | 0.05 | | | | |
| Optimizer | | | | AdamW | | | | |
| Batch size | | | | 32 | | | | |
| Warmup Steps | | | | 0 | | | | |
| Epochs | | | | 1 | | | | |
| Where | | | q, k, v, o, gate, up, down | | | | | |

We can now show that $\|X\|_F$ is small with high probability. Let the failure probability be $\delta$. By setting the bound from the previous step to $\delta$, we can solve for $t$:

$$\delta = 2r^2 \exp\left(\frac{-t^2 d_{\text{in}}}{18}\right) \implies t = \sqrt{\frac{18\log(2r^2/\delta)}{d_{\text{in}}}}. \tag{12}$$

With probability at least $1 - \delta$, we have $\max_{k,l} |X_{kl}| \leq t$. This allows us to bound the Frobenius norm of $X$:

$$\|X\|_F^2 = \sum_{k,l=1}^{r} |X_{kl}|^2 \leq r^2 (\max_{k,l} |X_{kl}|)^2 \leq r^2 t^2. \tag{13}$$

Thus, with probability at least $1 - \delta$:

$$\|X\|_F \leq r \cdot t = r\sqrt{\frac{18\log(2r^2/\delta)}{d_{\text{in}}}} = O\left(r\sqrt{\frac{\log r}{d_{\text{in}}}}\right). \tag{14}$$

Since $r \ll d_{\text{in}}$, the term $\|X\|_F \to 0$ as $d_{\text{in}} \to \infty$. This shows that $X$ converges to the zero matrix in probability.

Finally, we bound the magnitude of the original inner product using the Cauchy-Schwarz inequality for the Frobenius inner product and the sub-multiplicative property of the Frobenius norm:

$$\begin{aligned} |\langle \Delta_s, \Delta_t \rangle_F| = |\text{Tr}(\tilde{B}_s^\top X \tilde{B}_t)| &= |\langle \tilde{B}_s, X\tilde{B}_t \rangle_F| \\ &\leq \|\tilde{B}_s\|_F \|X\tilde{B}_t\|_F \\ &\leq \|\tilde{B}_s\|_F \|X\|_F \|\tilde{B}_t\|_F. \end{aligned} \tag{15}$$

The norms $\|\tilde{B}_s\|_F$ and $\|\tilde{B}_t\|_F$ are finite, as determined by the trained adapters. Since we have shown that $\|X\|_F \to 0$ in probability, the entire expression must also converge to 0 in probability. $\qquad\square$

## D  Hyperparameter Settings

We summarize the hyperparameter settings used for LoRI in Tables 4, 5, 6, and 7. These include settings for different tasks (NLU, math, code, safety), adapter variants (LoRI-D, LoRI-S), base models (Llama-3-8B and Mistral-7B), and ranks (32 and 64).

For the merging experiments, the hyperparameter settings for merging four adapters are provided in Tables 8 and 9, while those for merging three adapters are provided in Table 10.

Table 5: Hyperparameter settings for LoRI on the math dataset GSM8K.

| Method | LoRI-D | LoRI-S | LoRI-D | LoRI-S | LoRI-D | LoRI-S | LoRI-D | LoRI-S |
|---|---|---|---|---|---|---|---|---|
| Base Model | Llama-3 | Llama-3 | Llama-3 | Llama-3 | Mistral | Mistral | Mistral | Mistral |
| Rank $r$ | 32 | 32 | 64 | 64 | 32 | 32 | 64 | 64 |
| $\alpha$ | 64 | 64 | 128 | 128 | 64 | 64 | 32 | 64 |
| Sparsity Ratio | 0 | 0.9 | 0 | 0.9 | 0 | 0.9 | 0 | 0.9 |
| Learning Rate | 5e-5 | 5e-4 | 5e-5 | 1e-3 | 5e-5 | 5e-4 | 1e-4 | 5e-4 |
| Dropout | | | | 0.05 | | | | |
| Optimizer | | | | AdamW | | | | |
| Batch size | | | | 32 | | | | |
| Warmup Steps | | | | 0 | | | | |
| Epochs | | | | 3 | | | | |
| Where | | | | q, k, v, o, gate, up, down | | | | |

Table 6: Hyperparameter settings for LoRI on the code dataset CodeAlpaca.

| Method | LoRI-D | LoRI-S | LoRI-D | LoRI-S | LoRI-D | LoRI-S | LoRI-D | LoRI-S |
|---|---|---|---|---|---|---|---|---|
| Base Model | Llama-3 | Llama-3 | Llama-3 | Llama-3 | Mistral | Mistral | Mistral | Mistral |
| Rank $r$ | 32 | 32 | 64 | 64 | 32 | 32 | 64 | 64 |
| $\alpha$ | 64 | 64 | 128 | 128 | 64 | 64 | 128 | 128 |
| Sparsity Ratio | 0 | 0.9 | 0 | 0.9 | 0 | 0.9 | 0 | 0.9 |
| Learning Rate | 5e-5 | 5e-4 | 1e-5 | 1e-4 | 5e-5 | 5e-4 | 1e-5 | 1e-4 |
| Dropout | | | | 0.05 | | | | |
| Optimizer | | | | AdamW | | | | |
| Batch size | | | | 32 | | | | |
| Warmup Steps | | | | 0 | | | | |
| Epochs | | | | 2 | | | | |
| Where | | | | q, k, v, o, gate, up, down | | | | |

Table 7: Hyperparameter settings for LoRI on the safety dataset Saferpaca.

| Method | LoRI-D | LoRI-S | LoRI-D | LoRI-S | LoRI-D | LoRI-S | LoRI-D | LoRI-S |
|---|---|---|---|---|---|---|---|---|
| Base Model | Llama-3 | Llama-3 | Llama-3 | Llama-3 | Mistral | Mistral | Mistral | Mistral |
| Rank $r$ | 32 | 32 | 64 | 64 | 32 | 32 | 64 | 64 |
| $\alpha$ | 64 | 64 | 128 | 128 | 64 | 64 | 128 | 128 |
| Sparsity Ratio | 0 | 0.9 | 0 | 0.9 | 0 | 0.9 | 0 | 0.9 |
| Learning Rate | 5e-5 | 5e-4 | 1e-5 | 1e-4 | 5e-5 | 5e-4 | 1e-5 | 1e-4 |
| Dropout | | | | 0.05 | | | | |
| Optimizer | | | | AdamW | | | | |
| Batch size | | | | 32 | | | | |
| Warmup Steps | | | | 0 | | | | |
| Epochs | | | | 1 | | | | |
| Where | | | | q, k, v, o, gate, up, down | | | | |

Table 8: Hyperparameter settings for merging four adapters using Llama-3-8B.

| Adaptation Merging | LoRA Concat | LoRA Linear | LoRA Magnitude | LoRA TIES | LoRA DARE | LoRI-D Concat | LoRI-D Linear | LoRI-S Concat | LoRI-S Linear |
|---|---|---|---|---|---|---|---|---|---|
| Base Model | Llama-3 | Llama-3 | Llama-3 | Llama-3 | Llama-3 | Llama-3 | Llama-3 | Llama-3 | Llama-3 |
| Weights | 0.4 | 0.4 | 0.4 | 0.4 | 0.4 | 0.4 | 0.4 | 0.3 | 0.3 |
| Density | - | - | 0.3 | 0.7 | 0.7 | - | - | - | - |

Table 9: Hyperparameter settings for merging four adapters using Mistral-7B.

| Adaptation Merging | LoRA Concat | LoRA Linear | LoRA Magnitude | LoRA TIES | LoRA DARE | LoRI-D Concat | LoRI-D Linear | LoRI-S Concat | LoRI-S Linear |
|---|---|---|---|---|---|---|---|---|---|
| Base Model | Mistral | Mistral | Mistral | Mistral | Mistral | Mistral | Mistral | Mistral | Mistral |
| Weights | 0.4 | 0.4 | 0.4 | 0.4 | 0.4 | 0.4 | 0.4 | 0.3 | 0.3 |
| Density | - | - | 0.3 | 0.7 | 0.7 | - | - | - | - |

Table 10: Hyperparameter settings for merging three adapters using Llama-3-8B.

| Adaptation Merging | LoRA Concat | LoRA Linear | LoRA Magnitude | LoRA TIES | LoRA DARE | LoRI-D Concat | LoRI-D Linear | LoRI-S Concat | LoRI-S Linear |
|---|---|---|---|---|---|---|---|---|---|
| Base Model Weights | Llama-3 0.5 | Llama-3 0.5 | Llama-3 0.5 | Llama-3 0.5 | Llama-3 0.5 | Llama-3 0.5 | Llama-3 0.5 | Llama-3 0.4 | Llama-3 0.4 |
| Density | - | - | 0.3 | 0.7 | 0.7 | - | - | - | - |

Table 11: Performance comparison of different adaptation methods on eight NLU benchmarks using Llama-3 with $r = 32$. **Bold** indicates the best-performing method, and underline indicates the second-best.

| Method | # Params (%) | BoolQ | PIQA | SIQA | ARC-c | ARC-e | OBQA | HellaS | WinoG | Avg. |
|---|---|---|---|---|---|---|---|---|---|---|
| FFT | 8.03G (100%) | 73.8 | 86.8 | 77.6 | 76.7 | 87.6 | 84.1 | 93.2 | 85.1 | 83.1 |
| LoRA | 84M (1.03%) | 76.3 | **89.8** | 82.7 | 83.4 | 91.7 | 88.4 | 95.8 | **88.7** | 87.1 |
| VeRA | 1.38M (0.02%) | 64.4 | 81.8 | 62.6 | 67.3 | 85.7 | 60.9 | 78.5 | 56.9 | 69.8 |
| IA3 | 1.70M (0.02%) | 68.6 | 84.8 | 74.5 | 77.6 | 89.4 | 75.7 | 90.6 | 75.0 | 79.5 |
| LoRA-FA | 44M (0.54%) | 74.0 | 89.6 | **83.3** | 83.8 | 93.4 | **88.6** | **96.1** | 87.4 | 87.0 |
| AdaLoRA | 84M (1.03%) | 75.6 | 89.2 | 82.4 | 83.1 | 91.0 | 87.8 | 94.4 | 87.6 | 86.4 |
| rsLoRA | 84M (1.03%) | 72.8 | 84.8 | 78.8 | 76.0 | 87.0 | 85.0 | 91.0 | 82.8 | 82.3 |
| PiSSA | 84M (1.03%) | 68.1 | 84.4 | 78.2 | 75.1 | 85.1 | 82.8 | 89.3 | 82.8 | 80.7 |
| LoRA+ | 84M (1.03%) | 67.0 | 80.3 | 78.5 | 70.1 | 82.3 | 81.5 | 88.9 | 79.7 | 78.5 |
| DoRA | 85M (1.05%) | 75.9 | **89.8** | 82.7 | 83.5 | 93.2 | 87.9 | 95.3 | 88.2 | 87.1 |
| LoRI-D | 44M (0.54%) | **76.4** | 89.0 | 82.7 | **84.2** | **93.6** | 88.5 | 95.9 | 87.9 | **87.3** |
| LoRI-S | 4.4M (0.05%) | 75.2 | 89.2 | 82.8 | 83.8 | 92.6 | 88.4 | 95.2 | 87.5 | 86.8 |

# E  Additional Experimental Results

## E.1  Comparison with Additional PEFT Methods

To provide a comprehensive benchmark, we evaluate LoRI against several widely adopted parameter-efficient fine-tuning (PEFT) methods, including VeRA (Kopiczko et al., 2023), IA3 (Liu et al., 2022), LoRA-FA (Zhang et al., 2023b), AdaLoRA (Zhang et al., 2023d), rsLoRA (Kalajdzievski, 2023), PiSSA (Meng et al., 2024), LoRA+ (Hayou et al., 2024), and DoRA (Liu et al., 2024). The results, presented in Tables 11 and 12, demonstrate that our proposed methods are highly effective.

LoRI-D, which uses 44M trainable parameters (0.54% of the full model and half of LoRA's), consistently achieves state-of-the-art performance, particularly on NLU and code generation benchmarks. LoRI-S, despite its aggressive sparsity (0.05% of the full model and 5% of LoRA's), remains highly competitive and often surpasses other PEFT methods. While VeRA and IA3 are more parameter-efficient, their performance is substantially lower than LoRI-S. Despite this efficiency, LoRI-D and LoRI-S deliver comparable – and often superior – performance across NLU, math, code, and safety domains. These results underscore two key insights: (1) effective adaptation does not require updating the projection matrices $A$, as demonstrated by LoRI-D; and (2) the matrices $B$ contains significant redundancy that can be effectively pruned, as shown by LoRI-S.

## E.2  Results with Rank $r = 64$

We evaluate several adaptation methods using a higher adapter rank of $r = 64$ across a diverse set of tasks. This allows for more expressive adapter representations while still maintaining efficiency compared to full fine-tuning. Table 13 presents performance on eight natural language understanding (NLU) benchmarks, while Table 14 includes results on GSM8K (math), HumanEval (code), and HEx-PHI (safety). Across Llama-3 and Mistral models, LoRI-D and LoRI-S consistently perform competitively, often outperforming larger adapter methods like LoRA and DoRA, while using fewer parameters.

Table 12: Performance comparison of different adaptation methods on GSM8K (math), HumanEval (code), and HEx-PHI (safety) benchmarks using Llama-3 with $r = 32$. **Bold** indicates the best-performing method, and underline indicates the second-best.

| Method | # Params (%) | GSM8K | HumanEval | | | HEx-PHI |
| | | | Pass@1 | Pass@5 | Pass@10 | |
|---|---|---|---|---|---|---|
| FFT | 8.03G (100%) | 58.8 | 30.5 | 39.3 | 41.7 | 94.8 |
| LoRA | 84M (1.03%) | 64.4 | 34.7 | 46.4 | 50.8 | 91.6 |
| VeRA | 1.38M (0.02%) | 30.6 | 32.4 | 45.1 | 50.9 | 74.7 |
| IA3 | 1.70M (0.02%) | 48.0 | 32.7 | 45.6 | 51.5 | 85.4 |
| LoRA-FA | 44M (0.54%) | 64.8 | 42.9 | 57.5 | **64.2** | 94.1 |
| AdaLoRA | 84M (1.03%) | 63.3 | 33.5 | 45.0 | 49.4 | 91.9 |
| rsLoRA | 84M (1.03%) | 61.3 | 28.4 | 35.5 | 38.3 | 98.1 |
| PiSSA | 84M (1.03%) | 61.3 | 32.0 | 40.3 | 43.3 | 97.8 |
| LoRA+ | 84M (1.03%) | 61.7 | 33.0 | 42.7 | 46.0 | **98.8** |
| DoRA | 85M (1.05%) | **65.4** | 33.1 | 44.0 | 48.6 | 93.6 |
| LoRI-D | 44M (0.54%) | 63.2 | **43.2** | **57.6** | 63.2 | 92.8 |
| LoRI-S | 4.4M (0.05%) | 62.7 | 41.3 | 54.4 | 59.6 | 93.8 |

Table 13: Performance comparison of different adaptation methods on eight natural language understanding (NLU) benchmarks using Llama-3 and Mistral with $r = 64$. **Bold** indicates the best-performing method, and underline indicates the second-best.

| Method | # Params (%) | BoolQ | PIQA | SIQA | ARC-c | ARC-e | OBQA | HellaS | WinoG | Avg. |
|---|---|---|---|---|---|---|---|---|---|---|
| *Llama-3-8B* | | | | | | | | | | |
| FFT | 8.03G (100%) | 73.8 | 86.8 | 77.6 | 76.7 | 87.6 | 84.1 | 93.2 | 85.1 | 83.1 |
| LoRA | 168M (2.05%) | 75.2 | 89.0 | 81.2 | 82.3 | 92.4 | **89.1** | 95.3 | **88.2** | 86.6 |
| DoRA | 169M (2.06%) | 76.4 | 89.0 | 82.0 | 82.6 | 92.3 | 87.5 | 95.1 | 87.3 | 86.5 |
| LoRI-D | 88M (1.07%) | 75.8 | **90.4** | **82.7** | 83.3 | 92.6 | 88.6 | 95.9 | 87.4 | **87.1** |
| LoRI-S | 8.8M (0.11%) | **76.5** | 90.2 | 81.9 | **83.5** | **93.8** | 87.5 | **96.2** | 87.2 | **87.1** |
| *Mistral-7B* | | | | | | | | | | |
| FFT | 7.24G (100%) | 74.1 | 84.6 | 78.0 | 79.3 | 90.5 | 88.4 | 94.4 | 83.5 | 84.1 |
| LoRA | 168M (2.26%) | **77.4** | 90.2 | 83.5 | **84.0** | **93.0** | 89.3 | 95.6 | 89.4 | **87.8** |
| DoRA | 169M (2.28%) | 76.0 | 90.6 | 83.5 | 83.3 | 92.8 | 89.6 | 95.7 | 87.6 | 87.4 |
| LoRI-D | 88M (1.18%) | 75.9 | **90.7** | **83.7** | 82.0 | 92.1 | **90.0** | **96.4** | 87.8 | 87.3 |
| LoRI-S | 8.8M (0.12%) | 74.2 | **90.7** | 83.5 | 83.0 | 92.6 | 89.5 | 95.8 | **89.5** | 87.3 |

Table 14: Performance comparison of different adaptation methods on GSM8K (math), HumanEval (code), and HEx-PHI (safety) benchmarks using Llama-3 and Mistral with $r = 64$. **Bold** indicates the best-performing method, and underline indicates the second-best.

| Method | # Params (%) | GSM8K | HumanEval | | | HEx-PHI |
| | | | Pass@1 | Pass@5 | Pass@10 | |
|---|---|---|---|---|---|---|
| *Llama-3-8B* | | | | | | |
| FFT | 8.03G (100%) | 58.8 | 30.5 | 39.3 | 41.7 | 94.8 |
| LoRA | 168M (2.05%) | **63.9** | 38.6 | 52.9 | 59.2 | 94.1 |
| DoRA | 169M (2.06%) | 63.8 | 39.4 | 53.6 | 59.7 | 93.4 |
| LoRI-D | 88M (1.07%) | 63.8 | 41.9 | 55.4 | 60.3 | **96.6** |
| LoRI-S | 8.8M (0.11%) | 61.8 | **44.1** | **57.4** | **62.4** | 96.3 |
| *Mistral-7B* | | | | | | |
| FFT | 7.24G (100%) | 55.5 | 30.5 | 39.3 | 41.7 | 94.1 |
| LoRA | 168M (2.26%) | 56.7 | **33.9** | 43.1 | 46.9 | 95.9 |
| DoRA | 169M (2.28%) | 57.8 | 32.9 | 43.3 | 47.2 | **96.6** |
| LoRI-D | 88M (1.18%) | 58.2 | 33.3 | **43.6** | **47.3** | 90.9 |
| LoRI-S | 8.8M (0.12%) | **58.4** | 32.1 | 42.2 | 46.3 | 93.4 |

Table 15: Comparison of merging methods for combining four adapters, evaluated on their respective benchmarks. The best-performing single-task adapter, LoRI-D, is used as the single-task baseline. Results for NLU are averaged over eight tasks. Base model: Mistral-7B, rank $r = 32$. **Bold** indicates the best-performing method, and underline indicates the second-best.

| Merging | Adaptation | NLU | GSM8K | HumanEval | | | HEx-PHI |
| | | | | Pass@1 | Pass@5 | Pass@10 | |
|---|---|---|---|---|---|---|---|
| Single-Task | LoRI-D | 87.1 | 58.0 | 33.8 | 42.0 | 45.1 | 94.7 |
| Concat | LoRA | **82.5** | **52.4** | 32.3 | 40.8 | 44.1 | 75.6 |
| Linear | LoRA | 81.4 | 48.0 | 33.1 | 41.6 | 43.9 | 76.6 |
| Magnitude | LoRA | 77.5 | 42.7 | 32.7 | 41.8 | 45.6 | 80.9 |
| TIES | LoRA | 31.3 | 23.5 | 32.0 | 40.2 | 43.5 | 81.9 |
| DARE | LoRA | 76.1 | 43.0 | 32.0 | 41.0 | 44.6 | 83.4 |
| Concat | LoRI-D | 79.3 | **52.4** | 34.4 | **42.8** | 45.5 | **83.8** |
| Linear | LoRI-D | 78.1 | 50.5 | **35.2** | 42.7 | 45.5 | 79.7 |
| Concat | LoRI-S | 79.2 | 46.1 | 33.3 | 41.6 | **45.9** | 79.4 |
| Linear | LoRI-S | 75.5 | 40.3 | 28.8 | 36.0 | 39.6 | 83.1 |

Table 16: Comparison of merging methods for combining four adapters on eight NLU benchmarks. The best-performing single-task adapter, LoRI-D, is used as the single-task baseline. Base model: Llama-3-8B, rank $r = 32$. **Bold** indicates the best-performing method, and underline indicates the second-best.

| Merging | Adaptation | BoolQ | PIQA | SIQA | ARC-c | ARC-e | OBQA | HellaS | WinoG | Avg. |
|---|---|---|---|---|---|---|---|---|---|---|
| Single-Task | LoRI-D | 76.4 | 89.0 | 82.7 | 84.2 | 93.6 | 88.5 | 95.9 | 87.9 | 87.3 |
| Concat | LoRA | 73.9 | **89.1** | **81.1** | **81.4** | **92.4** | 83.0 | **94.4** | **84.5** | **85.0** |
| Linear | LoRA | 73.7 | 88.8 | **81.1** | 80.7 | 91.6 | **84.4** | 93.9 | 84.1 | 84.8 |
| Magnitude | LoRA | 72.0 | 87.1 | 76.8 | 79.4 | 91.7 | 81.5 | 90.4 | 76.4 | 81.9 |
| TIES | LoRA | 68.2 | 83.8 | 67.3 | 69.5 | 87.8 | 69.2 | 73.3 | 61.4 | 72.6 |
| DARE | LoRA | 70.7 | 85.0 | 74.1 | 77.5 | 90.7 | 76.6 | 86.8 | 71.0 | 79.1 |
| Concat | LoRI-D | **74.0** | 87.7 | 77.8 | 81.0 | **92.4** | 81.0 | 92.7 | 78.9 | 83.2 |
| Linear | LoRI-D | 73.7 | 87.7 | 76.7 | 80.3 | 92.1 | 80.1 | 92.0 | 77.7 | 82.5 |
| Concat | LoRI-S | 71.8 | 86.2 | 76.1 | 79.2 | 91.5 | 78.6 | 89.8 | 76.3 | 81.2 |
| Linear | LoRI-S | 70.7 | 85.3 | 75.1 | 78.0 | 90.8 | 75.0 | 86.5 | 71.3 | 79.1 |

## E.3   Merging Four Adapters

To support multi-task learning within a unified model, we study the merging of four task-specific adapters using various strategies. Table 15 reports results using Mistral-7B across a range of tasks. Additionally, Tables 16 and 17 break down the performance of NLU on individual benchmarks using Llama-3 and Mistral, respectively. We compare merging methods such as concatenated merging, linear merging, magnitude pruning, TIES, and DARE. LoRI-based approaches demonstrate strong performance and stability when merging multiple adapters.

## E.4   Merging Three Adapters

We further evaluate the merging of three adapters to understand performance when adapting to a smaller set of tasks. Tables 18 and 19 summarize the results for Llama-3 across different benchmarks. Similar to the four-task setting, LoRI-D remains a strong performer, often exceeding the performance of LoRA. These results highlight that LoRI-based methods are effective with varying levels of task diversity.

## E.5   Pruning-Based Merging Methods

Finally, we explore pruning-based merging methods, which aim to compress and combine multiple adapters by selectively retaining important weights. We focus on three methods: magnitude pruning, TIES, and DARE. Results are reported for merging both four-adapter

Table 17: Comparison of merging methods for combining four adapters on eight NLU benchmarks. The best-performing single-task adapter, LoRI-D, is used as the single-task baseline. Base model: Mistral-7B, rank $r = 32$. **Bold** indicates the best-performing method, and underline indicates the second-best.

| Merging | Adaptation | BoolQ | PIQA | SIQA | ARC-c | ARC-e | OBQA | HellaS | WinoG | Avg. |
|---|---|---|---|---|---|---|---|---|---|---|
| Single-Task | LoRI-D | 75.9 | 90.6 | 83.0 | 83.6 | 91.9 | 88.4 | 95.9 | 87.4 | 87.1 |
| Concat | LoRA | 69.0 | **88.0** | **78.1** | **79.9** | **90.9** | **84.2** | **92.4** | **77.8** | **82.5** |
| Linear | LoRA | 69.2 | 86.9 | 77.9 | 78.5 | 90.2 | 82.1 | 91.5 | 75.1 | 81.4 |
| Magnitude | LoRA | 68.7 | 84.9 | 74.4 | 75.9 | 89.1 | 77.5 | 85.6 | 64.1 | 77.5 |
| TIES | LoRA | 18.4 | 69.8 | 40.7 | 14.0 | 21.9 | 20.1 | 14.6 | 50.9 | 31.3 |
| DARE | LoRA | 69.4 | 84.3 | 73.1 | 74.2 | 88.9 | 74.3 | 82.6 | 61.8 | 76.1 |
| Concat | LoRI-D | 68.4 | 85.9 | 75.6 | 76.6 | 89.4 | 81.3 | 85.9 | 71.1 | 79.3 |
| Linear | LoRI-D | 66.3 | 86.0 | 74.9 | 75.3 | 88.9 | 80.8 | 85.0 | 68.0 | 78.1 |
| Concat | LoRI-S | **72.6** | 85.4 | 74.6 | 76.5 | 89.7 | 80.1 | 86.0 | 68.9 | 79.2 |
| Linear | LoRI-S | 67.6 | 83.8 | 72.0 | 73.0 | 88.3 | 74.6 | 80.9 | 64.3 | 75.5 |

Table 18: Comparison of merging methods for combining three adapters, evaluated on their respective benchmarks. The best-performing single-task adapter, LoRI-D, is used as the single-task baseline. Results for NLU are averaged over eight tasks. Base model: Llama-3-8B, rank $r = 32$. **Bold** indicates the best-performing method, and underline indicates the second-best.

| Merging | Adaptation | NLU | GSM8K | HumanEval | | |
| | | | | Pass@1 | Pass@5 | Pass@10 |
|---|---|---|---|---|---|---|
| Single-Task | LoRI-D | 87.3 | 63.2 | 43.2 | 57.6 | 63.2 |
| Concat | LoRA | **86.4** | 54.5 | 13.0 | 19.8 | 21.8 |
| Linear | LoRA | 86.1 | 51.9 | 8.8 | 14.5 | 16.7 |
| Magnitude | LoRA | 83.8 | 52.0 | 23.3 | 37.4 | 43.0 |
| TIES | LoRA | 79.4 | 26.9 | 36.3 | 48.7 | 53.7 |
| DARE | LoRA | 81.1 | 53.3 | 36.0 | 49.5 | 53.9 |
| Concat | LoRI-D | 84.8 | **59.6** | **41.5** | **56.4** | **61.6** |
| Linear | LoRI-D | 84.6 | 57.6 | 38.3 | 51.6 | 56.8 |
| Concat | LoRI-S | 83.3 | 51.8 | 31.2 | 44.6 | 49.8 |
| Linear | LoRI-S | 81.0 | 41.7 | 26.6 | 40.0 | 44.6 |

Table 19: Comparison of merging methods for combining three adapters on eight NLU benchmarks. The best-performing single-task adapter, LoRI-D, is used as the single-task baseline. Base model: Llama-3-8B, rank $r = 32$. **Bold** indicates the best-performing method, and underline indicates the second-best.

| Merging | Adaptation | BoolQ | PIQA | SIQA | ARC-c | ARC-e | OBQA | HellaS | WinoG | Avg. |
|---|---|---|---|---|---|---|---|---|---|---|
| Single-Task | LoRI-D | 76.4 | 89.0 | 82.7 | 84.2 | 93.6 | 88.5 | 95.9 | 87.9 | 87.3 |
| Concat | LoRA | **74.7** | **89.6** | **81.8** | **82.9** | **93.7** | **86.2** | **95.8** | 86.8 | **86.4** |
| Linear | LoRA | 73.9 | **89.6** | 81.4 | 81.9 | 93.5 | 85.5 | 95.6 | **87.1** | 86.1 |
| Magnitude | LoRA | 72.2 | 87.2 | 78.9 | 81.2 | 92.2 | 83.2 | 93.0 | 82.4 | 83.8 |
| TIES | LoRA | 69.5 | 84.8 | 74.0 | 78.4 | 91.2 | 77.4 | 88.8 | 71.4 | 79.4 |
| DARE | LoRA | 71.0 | 85.6 | 75.8 | 79.5 | 91.0 | 78.8 | 90.7 | 76.2 | 81.1 |
| Concat | LoRI-D | 73.8 | 89.0 | 79.8 | 81.0 | 93.0 | 83.0 | 94.6 | 84.0 | 84.8 |
| Linear | LoRI-D | 74.1 | 88.4 | 80.2 | 81.3 | 92.9 | 82.1 | 94.1 | 83.6 | 84.6 |
| Concat | LoRI-S | 70.3 | 87.2 | 79.1 | 80.8 | 92.4 | 82.1 | 93.2 | 81.3 | 83.3 |
| Linear | LoRI-S | 61.5 | 86.4 | 78.0 | 79.5 | 91.7 | 80.8 | 91.3 | 78.5 | 81.0 |

Table 20: Comparison of magnitude pruning, TIES, and DARE for combining four adapters, evaluated on their respective benchmarks. The best-performing single-task adapter, LoRI-D, is used as the single-task baseline. Results for NLU are averaged over eight tasks. Base model: Llama-3-8B, rank $r = 32$. **Bold** indicates the best-performing method within each group.

| Merging | Adaptation | NLU | GSM8K | HumanEval | | | HEx-PHI |
| | | | | Pass@1 | Pass@5 | Pass@10 | |
|---|---|---|---|---|---|---|---|
| Single-Task | LoRI-D | 87.3 | 63.2 | 43.2 | 57.6 | 63.2 | 92.8 |
| Magnitude | LoRA | 81.9 | 50.3 | 24.1 | 36.7 | 42.4 | 74.4 |
| Magnitude | LoRI-D | **84.3** | **50.5** | **33.3** | **45.2** | **51.4** | **85.9** |
| Magnitude | LoRI-S | 76.4 | 35.2 | 25.2 | 36.5 | 41.0 | 68.4 |
| TIES | LoRA | 72.6 | 24.0 | 32.5 | 46.3 | 51.7 | 77.8 |
| TIES | LoRI-D | **79.1** | **38.0** | **40.3** | **54.6** | **59.8** | **85.3** |
| TIES | LoRI-S | 70.4 | 25.9 | 34.6 | 48.4 | 53.2 | 77.8 |
| DARE | LoRA | 79.1 | 48.9 | 34.1 | 48.7 | 53.5 | 74.1 |
| DARE | LoRI-D | **83.4** | **52.0** | **35.4** | **51.3** | **57.8** | **81.9** |
| DARE | LoRI-S | 73.4 | 27.2 | 34.8 | 48.1 | 53.5 | 75.3 |

Table 21: Comparison of magnitude pruning, TIES, and DARE for combining four adapters, evaluated on their respective benchmarks. The best-performing single-task adapter, LoRI-D, is used as the single-task baseline. Results for NLU are averaged over eight tasks. Base model: Mistral-7B, rank $r = 32$. **Bold** indicates the best-performing method within each group.

| Merging | Adaptation | NLU | GSM8K | HumanEval | | | HEx-PHI |
| | | | | Pass@1 | Pass@5 | Pass@10 | |
|---|---|---|---|---|---|---|---|
| Single-Task | LoRI-D | 87.1 | 58.0 | 33.8 | 42.0 | 45.1 | 94.7 |
| Magnitude | LoRA | **77.5** | **42.7** | **32.7** | **41.8** | **45.6** | **80.9** |
| Magnitude | LoRI-D | 76.0 | 41.5 | 29.0 | 36.0 | 38.7 | 79.4 |
| Magnitude | LoRI-S | 70.5 | 32.4 | 28.1 | 36.1 | 39.3 | 77.5 |
| TIES | LoRA | 31.3 | 23.5 | 32.0 | 40.2 | 43.5 | **81.9** |
| TIES | LoRI-D | 65.0 | **45.4** | **35.3** | **44.5** | **47.8** | 68.4 |
| TIES | LoRI-S | **67.8** | 32.9 | 28.6 | 37.2 | 40.8 | 78.4 |
| DARE | LoRA | 76.1 | **43.0** | **32.0** | **41.0** | 44.6 | 83.4 |
| DARE | LoRI-D | **76.2** | 42.3 | 29.2 | 37.1 | 40.7 | **89.1** |
| DARE | LoRI-S | 71.9 | 34.3 | 29.2 | 40.5 | **44.9** | 85.0 |

(Tables 20 and 21) and three-adapter (Table 22) settings, using Llama-3 and Mistral as base models. LoRI-D consistently achieves strong performance across all pruning-based merging methods. However, the performance of LoRI-S is somewhat lower in these settings. This is because pruning-based methods operate on the dense $A$ matrices but not on the sparse $B$ matrices. This mismatch leads to an inconsistent pruning scheme, which can result in a loss of effectiveness.

## F  Additional Ablation Studies

Figure 5 presents GSM8K accuracy across a grid of sparsity ratios and learning rates using Mistral-7B with rank $r = 64$. We observe that sparse adapters require larger learning rates to train effectively. In particular, models with high sparsity (e.g., above 70%) perform best with a learning rate of $10^{-4}$ or higher. This suggests that stronger optimization is necessary to compensate for limited capacity in sparse adapters.

In Figure 6, we analyze how sparsity is distributed across layers and projections when enforcing 90% global sparsity on GSM8K. We find that feedforward (FFN) projections tend to retain more parameters – i.e., they exhibit lower sparsity – than self-attention projections.

Table 22: Comparison of magnitude pruning, TIES, and DARE for combining three adapters, evaluated on their respective benchmarks. The best-performing single-task adapter, LoRI-D, is used as the single-task baseline. Results for NLU are averaged over eight tasks. Base model: Llama-3-8B, rank $r = 32$. **Bold** indicates the best-performing method within each group.

| Merging | Adaptation | NLU | GSM8K | HumanEval | | |
|---|---|---|---|---|---|---|
| | | | | Pass@1 | Pass@5 | Pass@10 |
| Single-Task | LoRI-D | 87.3 | 63.2 | 43.2 | 57.6 | 63.2 |
| Magnitude | LoRA | 83.8 | 52.0 | 23.3 | 37.4 | 43.0 |
| Magnitude | LoRI-D | **84.6** | **53.7** | **34.8** | **48.9** | **54.7** |
| Magnitude | LoRI-S | 77.8 | 36.6 | 25.5 | 38.8 | 43.8 |
| TIES | LoRA | 79.4 | 26.9 | 36.3 | 48.7 | 53.7 |
| TIES | LoRI-D | **82.1** | **42.2** | **39.2** | **52.7** | **57.7** |
| TIES | LoRI-S | 73.8 | 35.2 | 34.8 | 47.9 | 52.5 |
| DARE | LoRA | 81.1 | 53.3 | 36.0 | **49.5** | **53.9** |
| DARE | LoRI-D | **84.0** | **55.2** | 33.8 | 45.8 | 51.8 |
| DARE | LoRI-S | 75.3 | 36.6 | **36.2** | 48.9 | 53.4 |

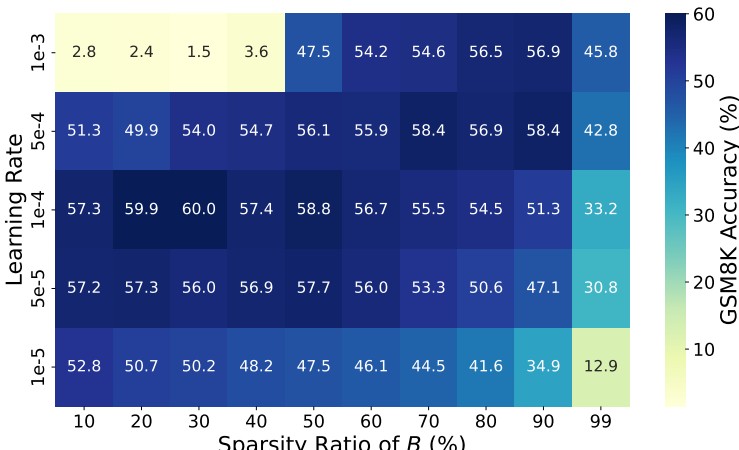

Figure 5: GSM8K accuracy under different sparsity ratios and learning rates. Base model: Mistral-7B, rank $r = 64$.

This indicates that FFN components are more critical for effective adaptation. Additionally, sparsity decreases toward the top of the network, suggesting that higher layers are more important for task-specific specialization.

Lastly, Figure 7 explores the effect of merging weights when combining three LoRI-S adapters using concatenated and linear merging. We find a noticeable trade-off between performance on code tasks and other domains (e.g., NLU and math). Higher merging weights can improve NLU performance but tend to degrade performance on code, highlighting the challenge of balancing generalization and specialization in multi-task settings.

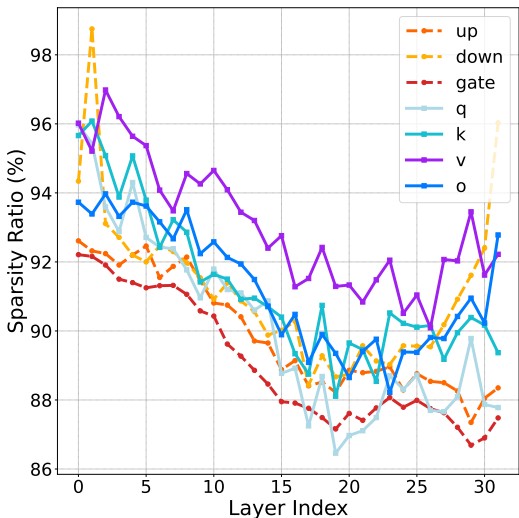

Figure 6: Sparsity ratios across layers and projections under a 90% sparsity on GSM8K. Base model: Llama-3-8B, rank $r = 32$.

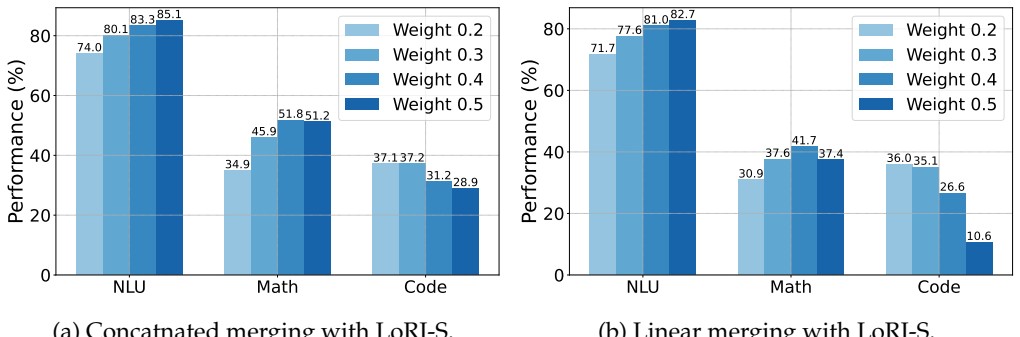

(a) Concatnated merging with LoRI-S.    (b) Linear merging with LoRI-S.

Figure 7: Ablation study on the effect of merging weights when combining three adapters. Base model: Llama-3-8B, rank $r = 32$.

