# OpenReview forum: "LoRI: Reducing Cross-Task Interference in Multi-Task Low-Rank Adaptation"
_colmweb.org/COLM/2025/Conference — COLM 2025_

### Official Review · Reviewer_nvBE · 2025-05-13

**Rating:** 7
**Confidence:** 4
**Ethics Flag:** 1

**Summary:**

This paper proposed LoRA with Reduced Interference (LoRI), a simple yet effective approach that freezes the projection matrices A as random projections and sparsifies the matrices B using task specific masks. I like this kinds of work which can be benefit for many downstream tasks.

Suggestions:

Huggingface has its public source code for load multiple adapters (https://huggingface.co/docs/peft/main/developer_guides/lora#load-adapters). Do you think it will be a good way to compare and what will be the advantages of your methods like adapter merging in Figure 1 (b) in this regards?
Can your method been applied to vision language model scenario? Since currently all tasks are been evaluating on text-based tasks.
There are some existing methods working on optimizing LoRA such as [1] and [2]. Do you think those methods can be considered as a baseline method or related work in your future work?
In the implementation details, the paper mentioned that the experiments are done using 8 Nvidia A5000 GPUs. My question is since this is an lora training. I am more interested in what will be the performance if model can be trained using a single A5000 GPU. In many real-world cases, the main reason we used lora is we did not have many GPUs so it will be helpful if the authors can provide an additional experiment to show if the performance will not degrade when we use less amount of GPUs.

[1] CoRA: Optimizing Low-Rank Adaptation with Common Subspace of Large Language Models, https://arxiv.org/abs/2409.02119

[2] LoRA-Pro: Are Low-Rank Adapters Properly Optimized? https://arxiv.org/abs/2407.18242

**Reasons To Accept:**

This paper has performed a large scale experiments which arcoss multiple tasks and demonstrated the effectiveness of the proposed method.

**Reasons To Reject:**

The github repo is not public and I believe it will be much help if the paper been accepted and public its code.
References:

---

> ### Author Response · Authors · 2025-06-01
>
> Thank you for your thoughtful review! Below, we address your questions in detail:
>
> > The github repo is not public and I believe it will be much help if the paper been accepted and public its code.
>
> Thank you for pointing this out! We have made the code available through an anonymous GitHub repository, which can be accessed here https://anonymous.4open.science/r/LoRI-28C6.
>
> > Hugging Face has public source code for loading multiple adapters. What are the advantages of your method?
>
> Our adapter merging is implemented using the `add_weighted_adapter` method from the PEFT library, leveraging the `cat` and `linear` merging methods already available there. The key advantage of LoRI lies in its reduction of interference during merging. By using independently initialized frozen A matrices, LoRI maps adapters into approximately orthogonal subspaces. This minimizes cross-task interference without requiring explicit regularization terms, unlike methods such as O-LoRA or AdaLoRA.
>
> > Can your method be applied to vision-language models?
>
> Yes, we believe LoRI can be extended to vision-language models. This is an exciting direction, and we consider it part of our future work.
>
> > There are some existing methods working on optimizing LoRA such as CoRA and LoRA-Pro.
>
> Thank you for pointing out these relevant works! We will make sure to include CoRA and LoRA-Pro in the related work section of a future version of our manuscript. We have added several widely adopted and representative PEFT baselines, including VeRA, IA3, LoRA-FA, AdaLoRA, rsLoRA, PiSSA, LoRA+, and DoRA. The updated results are available at our [anonymous GitHub repository](https://anonymous.4open.science/r/LoRI-28C6) and [anonymous Slides presentation](https://docs.google.com/presentation/d/e/2PACX-1vSzscuhb7-E9kFTfkS_9IMFTZu_-PPI4fOBIWoaPpfwzWMmhoNyrRTSv9I4UiFd3EdCO24nciys1D49/pub?start=false&loop=false&delayms=3000). Due to time constraints, we were only able to include a subset of baselines, but we plan to expand our comparisons in the future version.
>
> > What will be the performance if model can be trained using a single A5000 GPU?
>
> Thank you for raising this important point. We used 8 A5000 GPUs primarily due to cluster availability, not because LoRI requires that many GPUs. In fact, LoRI can be trained on a single A100 / H100 GPU. LoRI is also more memory-efficient than standard LoRA, as it freezes the A matrices, thereby reducing optimizer state memory and activation memory. Additionally, combining LoRI with quantization techniques like QLoRA can significantly lower the memory footprint, making it suitable for training on a single A5000. We consider LoRI + quantization a promising direction for future research.

---

> > ### Comment · Reviewer_nvBE · 2025-06-06
> >
> > Thanks for your reply. I raise up the raing score. I like the work and hope I can see this paper in the conference.

---

> > > ### Author Response · Authors · 2025-06-06
> > > **Thank You**
> > >
> > > Thank you for the update and for increasing the score. We truly appreciate it!

---

### Official Review · Reviewer_5RGh · 2025-05-13

**Rating:** 4
**Confidence:** 4
**Ethics Flag:** 1

**Summary:**

The paper introduces Low-Rank Adaptation with Reduced Interference (LoRI) a parameter-efficient training method suited for training multiple tasks and merging the resulting parameters for a combined inference. LoRI follows the architecture of LoRA with two low-rank matrices A and B, but instead of training both, LoRI keeps A freezer (randomly initialized) and imposes sparsity to B. The work further proposes combining the per-task trained delta parameters through concatenation and scaler weight learning. the authors show that this feature can also be leveraged for preserving the effectively of safety adapter in the inference.

The method is compared against the common baseline LoRA methods as well as merging methods, showing an overall good performance with much less trainable parameters.

**Reasons To Accept:**

- The paper is well written and clear.
- The method provides a practical solution for learning much less parameters (in comparison with standard LoRA methods) for a specific task.
- The

**Reasons To Reject:**

My main concern is the novelty of the method especially when considering the matureness and breadth of the studies on LoRA and PEFT. I believe the paper must provide a better analysis of the existing methods and shows - both in terms of method and experimental results - how the introduced approach different/better/preferred in comparison with other methods. The work in compared solely with "basic" baselines and the Related Work (which is appeared in Appendix) lacks relevant studies.

Related to the parameter-efficiency and simplicity in multi-task learning, this paper comes in my mind:
Frohmann, Markus, et al. "ScaLearn: Simple and Highly Parameter-Efficient Task Transfer by Learning to Scale." Findings of the Association for Computational Linguistics ACL 2024. 2024.

This work also seems relevant regarding safety preservation and bias mitigation:
Kumar, Deepak, et al. "Parameter-efficient Modularised Bias Mitigation via AdapterFusion." Proceedings of the 17th Conference of the European Chapter of the Association for Computational Linguistics. 2023.

I strongly encourage a deeper study in relevant studies and adding further baselines.

---

> ### Author Response · Authors · 2025-06-01
>
> Thank you for your thoughtful review! Below, we address your concerns in detail:
>
> > I strongly encourage a deeper study in relevant studies.
>
> Following your suggestion, we provide a discussion of related work relevant to LoRI. Given the breadth of research in the PEFT space, a comprehensive survey is infeasible, but we aim to highlight prior works that are most closely related to our work:
>
> - **Parameter Efficiency**: LoRI is motivated by reducing parameter redundancy in LoRA, which aligns with the goals of IA3, VeRA, and FourierFT. Unlike these methods, which introduce learnable vectors, we specifically target redundancy within LoRA and propose a new asymmetry: frozen A + sparse B.
> - **Frozen A Matrices**: The concept of asymmetric LoRA variants has been explored in LoRA-FA, AsymmetryLoRA, and HydraLoRA. However, to our knowledge, none of these combine a frozen A with a sparse B.
> - **LoRA Pruning**: Prior works such as Loraprune, LoRA-drop, RoseLoRA, and SoRA focus on pruning strategies, but do not perform targeted, asymmetric pruning solely on the B matrices. Other approaches like AdaLoRA and PiSSA prune based on singular values or vectors (via SVD), which is orthogonal to our direction.
>
> We also note that the aforementioned papers do not consider multi-task scenarios, which are a central focus of LoRI.
>
> - **Orthogonality in Multi-Task Learning**: While works like O-LoRA, AWD, and AdaLoRA enforce orthogonality via regularization, LoRI relies on the natural orthogonality of random frozen A matrices. Importantly, we leverage this property for clean adapter merging, which prior work does not address.
> - **Sparsity in Multi-Task Learning**: Methods such as SHiRA and LoTA explore sparsity in multi-task settings. However, our approach is distinct in that we use sparsity only in B to mitigate catastrophic forgetting in continual learning.
>
> We will revise and expand the related work section – particularly the PEFT part – in the future revision to provide a more comprehensive overview of prior work.
>
> > I strongly encourage adding further baselines.
>
> Thank you for the suggestion! We have added several widely adopted and representative PEFT methods as baselines, including **VeRA**, **IA3**, **LoRA-FA**, **AdaLoRA**, **rsLoRA**, **PiSSA**, **LoRA+**, and **DoRA**. We believe these additions provide a more comprehensive comparison and better highlight the strengths of our approach. The updated results can be found at our [anonymous GitHub repository](https://anonymous.4open.science/r/LoRI-28C6) and [anonymous Slides presentation](https://docs.google.com/presentation/d/e/2PACX-1vSzscuhb7-E9kFTfkS_9IMFTZu_-PPI4fOBIWoaPpfwzWMmhoNyrRTSv9I4UiFd3EdCO24nciys1D49/pub?start=false&loop=false&delayms=3000).
>
> As shown in the updated results, LoRI-D consistently achieves **the best performance** on NLU and code generation tasks, and performs on par with other methods on math and safety tasks, while using 44M trainable parameters (0.5% of full fine-tuning and 50% of LoRA). LoRI-S (0.05% parameters of full fine-tuning and 5% of LoRA), while slightly behind LoRI-D due to its more aggressive sparsity, remains highly competitive among existing PEFT methods. Notably, while VeRA and IA3 are more parameter-efficient – using 0.02% of the full model parameters – their performance is **significantly** lower than that of LoRI-S and other PEFT approaches.
>
> > Related to the parameter-efficiency and simplicity in multi-task learning, this paper comes in my mind: Frohmann, Markus, et al. "ScaLearn: Simple and Highly Parameter-Efficient Task Transfer by Learning to Scale." Findings of the Association for Computational Linguistics ACL 2024. 2024.
> >
> > This work also seems relevant regarding safety preservation and bias mitigation: Kumar, Deepak, et al. "Parameter-efficient Modularised Bias Mitigation via AdapterFusion." Proceedings of the 17th Conference of the European Chapter of the Association for Computational Linguistics. 2023.
>
> Thank you for sharing these relevant works! We will make sure to cite them and discuss their connections to our approach in the related work section of the future revision.

---

> ### Author Response · Authors · 2025-06-06
> **Looking Forward to Your Response**
>
> Dear Reviewer 5RGh,
>
> Thank you again for your thoughtful feedback and questions. We hope our response has addressed your concerns clearly.
>
> If you have any further questions or suggestions, we'd be happy to engage further. We would greatly appreciate any follow-up comments or clarification regarding your evaluation.
>
> Best regards,
>
> Authors

---

> > ### Author Response · Authors · 2025-06-09
> > **Looking Forward to Your Response**
> >
> > Dear Reviewer 5RGh,
> >
> > Thank you for your thoughtful review! We enacted your suggestion to add a range of further baselines and clarified our contributions, which we also shared with other reviewers. As a result of this, during the discussion period so far multiple reviewers have increased their scores.
> >
> > As the discussion period is coming to an end, we would greatly appreciate if you have any more helpful suggestions to improve our work. We look forward to hearing back from you.
> >
> > Best,
> >
> > Authors

---

### Official Review · Reviewer_cZht · 2025-05-14

**Rating:** 6
**Confidence:** 4
**Ethics Flag:** 1

**Summary:**

This paper proposes LoRI, a method that enhances parameter-efficient fine-tuning by freezing the projection matrices A and introducing sparsity into the matrices B. To mitigate cross-task interference during adapter merging, LoRI exploits the orthogonality of adapter subspaces. Additionally, it supports continual learning by leveraging sparsity to reduce catastrophic forgetting. Experimental results across a variety of tasks demonstrate the effectiveness of the proposed approach.

**Questions To Authors:**

- Before adapter merging and continual learning can be performed, the task-specific masks must first be obtained, which might introduce additional training overhead?

- Are there any discernible patterns in the structure or distribution of the sparse masks? Do they exhibit significant overlap across different tasks, or are they largely distinct?

**Reasons To Accept:**

- The study of methods to prevent parameter interference during LLM adaptation is an important topic and could benefit the community.

- The paper is generally easy to follow.

**Reasons To Reject:**

- The ideas presented in the paper are conceptually similar to existing work.

- Related works are not in the main body of the paper, but they are primarily included in the appendix.

---

> ### Author Response · Authors · 2025-06-01
>
> Thank you for your thoughtful review! Below, we address your concerns and questions in detail:
>
> > The ideas presented in the paper are conceptually similar to existing work.
>
> To the best of our knowledge, no prior work combines **frozen A** with **sparse B**. Additionally, we explicitly address **multi-task scenarios**, which are often overlooked in prior PEFT research. LoRI leverages **orthogonality** to reduce interference during adapter merging and **sparsity** to mitigate catastrophic forgetting in continual learning. We elaborate on the differences between our approach and prior works below:
>
> - **Parameter Efficiency**: LoRI is motivated by reducing parameter redundancy in LoRA, which aligns with the goals of IA3, VeRA, and FourierFT. Unlike these methods, which introduce learnable vectors, we specifically target redundancy within LoRA and propose a new asymmetry: frozen A + sparse B.
> - **Frozen A Matrices**: The concept of asymmetric LoRA variants has been explored in LoRA-FA, AsymmetryLoRA, and HydraLoRA. However, to our knowledge, none of these combine a frozen A with a sparse B.
> - **LoRA Pruning**: Prior works such as Loraprune, LoRA-drop, RoseLoRA, and SoRA focus on pruning strategies, but do not perform targeted, asymmetric pruning solely on the B matrices. Other approaches like AdaLoRA and PiSSA prune based on singular values or vectors (via SVD), which is orthogonal to our direction.
> - **Orthogonality in Multi-Task Learning**: While works like O-LoRA, AWD, and AdaLoRA enforce orthogonality via regularization, LoRI relies on the natural orthogonality of random frozen A matrices. Importantly, we leverage this property for clean adapter merging, which prior work does not address.
> - **Sparsity in Multi-Task Learning**: Methods such as SHiRA and LoTA explore sparsity in multi-task settings. However, our approach is distinct in that we use sparsity only in B to mitigate catastrophic forgetting in continual learning.
>
> We also welcome any pointers to specific works you find closely related to ours, and we would be happy to acknowledge and discuss them in future revisions.
>
> > Related works are not in the main body of the paper.
>
> Due to page constraints, we had to place the related work section in the appendix. If granted an additional page for the camera-ready version, we would include it in the main text for greater visibility.
>
> > The task-specific masks must first be obtained, which might introduce additional training overhead?
>
> Yes, calibration introduces some overhead, as it does in all pruning-based methods. The cost is comparable to a single fine-tuning pass. If needed, the number of calibration steps can be reduced to minimize overhead – at the cost of a small drop in performance, as shown in Figure 4(a). Importantly, masks are **calibrated only once per task** and can be reused in continual learning scenarios or future adaptations.
>
> > Are there any discernible patterns in sparse masks? Do they exhibit significant overlap across different tasks?
>
> We do not observe discernible patterns in sparse masks across tasks. In fact, task-specific masks are largely non-overlapping. At a 90% sparsity level, the average overlap is around 1%. Although enforcing strict non-overlap is possible – by preventing future updates from modifying previously used positions – we chose not to do so, as it would restrict the parameter space and reduce the reusability of calibrated masks.

---

> ### Author Response · Authors · 2025-06-06
> **Looking Forward to Your Response**
>
> Dear Reviewer cZht,
>
> Thank you again for your thoughtful feedback and questions. We hope our response has addressed your concerns clearly.
>
> If you have any further questions or suggestions, we'd be happy to engage further. We would greatly appreciate any follow-up comments or clarification regarding your evaluation.
>
> Best regards,
>
> Authors

---

> > ### Comment · Reviewer_cZht · 2025-06-08
> >
> > Apologies for the delayed response. The authors have addressed my concerns, and I have raised my score accordingly.

---

### Official Review · Reviewer_wCSw · 2025-05-16

**Rating:** 7
**Confidence:** 3
**Ethics Flag:** 1

**Summary:**

The authors propose LoRI, an adapter-based approach that freezes a shared random down-projection and sparsifies the up-projection. Across various tasks, including NLU, mathematical reasoning, code generation, and safety alignment tasks, it shows ~95 % parameter savings and effortless multi-task fusion compared to LoRA.

**Questions To Authors:**

1) Please clarify why the analysis doesn't include strong modern PEFT methods such as QLoRA, AdaLoRA or HyperLoRA?
2) In Table 1, the authors give LoRA almost twice the trainable parameters of LoRI-D (84 M vs 44 M), yet the average NLU score is separated by just 0.2 points (87.1 vs 87.3). With a fairer test, e.g. LoRA at rank 16 to match LoRI-D’s budget, there's every chance LoRA would equal or surpass it, but that experiment is missing.
3) For tasks that require a richer, learned subspace or very low-rank settings, LoRI can hit a ceiling that LoRA does not have because LoRA is allowed to tune both A and B.

**Reasons To Accept:**

1) It achieves extreme parameter and memory efficiency (>95 % smaller than LoRA) by freezing the down-projection and sparsifying the up-projection while maintaining competitive accuracy.
2) It minimises cross-task interference due to a shared random subspace, enabling straightforward adapter merging without re-training or complex heuristics.
3) It offers a simple hyperparameter-light pipeline (single mask-calibration step) that scales smoothly to hundreds of tasks across diverse domains.

**Reasons To Reject:**

1) The authors mainly repackage prior PEFT ideas via frozen random A (as in VeRA/O-LoRA) and sparsified B (as in AdaLoRA) rather than introducing a fundamentally new algorithmic insight.
2) The adapter’s updates are unstructured-sparse, which saves parameters on paper but often slows training and inference in practice because today’s GPU libraries are tuned for dense (or very structured-sparse) matrix multiplies.

---

> ### Author Response · Authors · 2025-06-01
>
> Thank you for your thoughtful review! Below we address your concerns and questions in detail:
>
> > It minimises cross-task interference due to a shared random subspace.
>
> We would like to clarify that in our adapter merging setup, we do **not** share the A matrix across adapters. Instead, each adapter uses a different random A matrix. These matrices are approximately orthogonal in high-dimensional space, which helps reduce interference. This orthogonality is achieved naturally, without the need for an explicit regularization term in the loss function used in prior works.
>
> > The authors mainly repackage prior PEFT ideas via frozen random A (as in VeRA/O-LoRA) and sparsified B (as in AdaLoRA) rather than introducing a fundamentally new algorithmic insight.
>
> We respectfully clarify that techniques used in LoRI are distinct from prior work. Specifically:
>
> - VeRA freezes both A and B but introduces trainable vectors; it does not use frozen A matrices in the way we do.
> - O-LoRA focuses on enforcing orthogonality between LoRAs via regularization but does not leverage frozen A matrices or sparse B matrices.
> - AdaLoRA performs pruning in the singular value domain, rather than applying sparsity to B.
>
> While some works such as LoRA-FA and HydraLoRA explore frozen A matrices, to the best of our knowledge, no existing method combines frozen A with sparse B, which is the core novelty of LoRI. For a broader comparison with relevant methods, we kindly refer you to [our response to Reviewer 5RGh](https://openreview.net/forum?id=b8cW86QcOD&noteId=2DUSqh7195).
>
> > It slows training and inference in practice because today’s GPU libraries are tuned for dense (or very structured-sparse) matrix multiplies.
>
> Thank you for raising this important point! We evaluated training throughput (measured in samples per second) on GSM8K using a single H100 GPU, and present the results below:
>
> |                                 | LoRA  | DoRA  | LoRI-D | LoRI-S |
> | :-----------------------------: | :---: | :---: | :----: | :----: |
> | Training Throughput (samples/s) | 17.45 | 10.10 | 17.87  | 16.84  |
>
> The results show that LoRI-D achieves slightly higher training throughput than LoRA and significantly outperforms DoRA. LoRI-S introduces a small overhead due to masking gradients during the backward pass, but the difference is modest.
>
> During inference, LoRI behaves similarly to LoRA. The low-rank updates are merged into the base model weights, meaning there is **no additional inference latency compared to full fine-tuning**. In practice, LoRI-D, LoRI-S, LoRA, and full fine-tuning all run inference at the same speed, as they all operate on the base model once merged.
>
> > Why doesn't the analysis include strong modern PEFT methods such as QLoRA, AdaLoRA, or HyperLoRA?
>
> We have included AdaLoRA as a baseline in our experiments. For results on additional widely adopted PEFT baselines, we kindly refer you to our [anonymous GitHub repository](https://anonymous.4open.science/r/LoRI-28C6) and [anonymous Slides presentation](https://docs.google.com/presentation/d/e/2PACX-1vSzscuhb7-E9kFTfkS_9IMFTZu_-PPI4fOBIWoaPpfwzWMmhoNyrRTSv9I4UiFd3EdCO24nciys1D49/pub?start=false&loop=false&delayms=3000).
>
> QLoRA is a quantization-based method focused on reducing memory usage. It typically introduces a slight performance degradation compared to LoRA. We consider LoRI + QLoRA a promising direction for future work to further reduce memory cost. HyperLoRA is tailored toward portrait synthesis and vision-related tasks. As our experiments are focused on text generation, we did not include HyperLoRA as a baseline.
>
> > LoRA at rank 16 vs LoRI-D at rank 32
>
> We have conducted experiments comparing LoRA at rank 16 and LoRI-D at rank 32, with results summarized below:
>
> |     Method      | NLU  | GSM8K | HumanEval | HEx-PHI |
> | :-------------: | :--: | :---: | :-------: | :-----: |
> |  LoRA (r = 16)  | 86.7 | 64.0  |   54.0    |  91.9   |
> |  LoRA (r = 32)  | 87.1 | 64.4  |   50.8    |  91.6   |
> | LoRI-D (r = 32) | 87.3 | 63.2  |   63.2    |  92.8   |
>
> These results show that LoRI-D at rank 32 outperforms both LoRA variants on NLU, code generation, and safety tasks, demonstrating its strong effectiveness.
>
> > LoRI can hit a ceiling that LoRA does not have.
>
> We agree that, in principle, LoRI-D and LoRI-S may have lower expressiveness than LoRA due to training fewer parameters. However, our aim is to **highlight the surprising extent of redundancy in LoRA**. Our results suggest that adaptation may not require updating A at all, and that B contains substantial redundancy. LoRI offers an efficient alternative while still achieving competitive performance across tasks.

---

> > ### Comment · Reviewer_wCSw · 2025-06-06
> >
> > I am increasing the score, as the authors have addressed all the concerns I raised.

---

> > > ### Author Response · Authors · 2025-06-06
> > > **Thank You**
> > >
> > > Thank you for the update and for increasing the score. We truly appreciate it!

---

### Official Review · Reviewer_nmP5 · 2025-05-16

**Rating:** 6
**Confidence:** 3
**Ethics Flag:** 1

**Summary:**

This paper introduces LoRI, which is based on LoRA but 1) leaves the A matrix unadapted and 2) learns elementwise sparse masks for B. By sharing the Gaussian-initialized A matrix the LoRA updates in a multi adaptor setting will be approximately orthogonal. And the learned sparsity further reduces inteference. Empirically LoRI has good single task performance, and also performs better under adapter merging and continual learning.

**Questions To Authors:**

Did the authors consider sensitivity analysis as a criterion for pruning B?

**Reasons To Accept:**

- Extensive results show that LoRI is competitive.

**Reasons To Reject:**

The sparsity matrix is created using the B matrix magnitudes. Justification is needed. For example, sensitivity analysis might be a more principled approach (relevant work in the area include [LQ-LoRA](https://openreview.net/forum?id=xw29VvOMmU) and [TaLoS](https://arxiv.org/abs/2504.02620). See also my questions to authors.

---

> ### Author Response · Authors · 2025-06-01
>
> Thank you for your thoughtful review! Below we address your questions in detail:
>
> > By sharing the Gaussian-initialized A matrix, the LoRA updates in a multi-adapter setting will be approximately orthogonal.
>
> We would like to clarify that in our adapter merging setup, we do **not** share the A matrix across adapters. Instead, each adapter uses a different random A matrix. These matrices are approximately orthogonal in high-dimensional space, which helps reduce interference. This orthogonality is achieved naturally, without the need for an explicit regularization term in the loss function used in prior works.
>
> > Sensitivity analysis might be a more principled approach for pruning. Did the authors consider sensitivity analysis as a criterion for pruning B?
>
> We chose magnitude-based masking due to its simplicity, efficiency, and strong empirical performance. Sensitivity-based approaches like those used in LQ-LoRA and TaLoS rely on the Fisher Information Matrix. To assess the effectiveness of different pruning strategies, we conducted experiments on GSM8K with LoRI-S using Llama-3-8B, comparing top-k magnitude (used in our method), Fisher score $(\nabla_B \mathcal{L})^2$, and SNIP score $|B \cdot \nabla_B \mathcal{L}|$. The results are as follows:
>
> |     Criterion      | Top-k Magnitude | Fisher Score | SNIP Score |
> | :----------------: | :-------------: | :----------: | :--------: |
> | GSM8K Accuracy (%) |      62.7       |     58.7     |    60.8    |
>
> The results show that **top-k magnitude pruning** yields the best performance. We attribute this to the fact that gradient-based measures capture local sensitivity at a particular training step, whereas magnitude reflects the cumulative importance of parameters over the entire fine-tuning process.

---

> > ### Comment · Reviewer_nmP5 · 2025-06-07
> > **Thanks for the response!**
> >
> > I really appreciate the additional comparison against Fisher information and SNIP, and will keep my score.

---

### Author Response · Authors · 2025-06-01
**Global Response**

We sincerely thank all the reviewers for their time and thoughtful feedback. In this work, our goal is to demonstrate that extensive parameter counts are not necessary to unlock the capabilities of base models during LLM fine-tuning. We also propose an efficient solution for interference-reduced adapter merging and continual learning. Our key contributions are summarized below:

- We introduce LoRI, a method that combines *frozen A matrices* with *sparse B matrices*, revealing that up to **95%** of LoRA parameters are redundant. We also demonstrate redundancy at the layer and projection levels in LoRA adapters (shown in Figure 4(b)).
- To reduce interference during **adapter merging**, we exploit the **orthogonality** of frozen random A matrices, avoiding the need for additional regularization for orthogonality.
- To reduce interference during **continual learning**, we leverage **sparsity** in B to isolate updates across tasks, enabling lightweight safety adapters with minimal catastrophic forgetting of safety alignment.

In response to reviewer suggestions, we have released an anonymous GitHub repository to support reproducibility: https://anonymous.4open.science/r/LoRI-28C6. We have also expanded our experimental comparisons to include widely adopted and representative PEFT baselines, such as **VeRA**, **IA3**, **LoRA-FA**, **AdaLoRA**, **rsLoRA**, **PiSSA**, **LoRA+**, and **DoRA**. Detailed results are available in our [anonymous GitHub repository](https://anonymous.4open.science/r/LoRI-28C6), and visualizations of these results are provided in this [anonymous Slides presentation](https://docs.google.com/presentation/d/e/2PACX-1vSzscuhb7-E9kFTfkS_9IMFTZu_-PPI4fOBIWoaPpfwzWMmhoNyrRTSv9I4UiFd3EdCO24nciys1D49/pub?start=false&loop=false&delayms=3000).

As shown in the updated results, LoRI-D consistently achieves **the best performance** on NLU and code generation tasks, and performs on par with other methods on math and safety tasks, while using 44M trainable parameters (**0.5%** of full fine-tuning and **50%** of LoRA). LoRI-S (**0.05%** parameters of full fine-tuning and **5%** of LoRA), remains highly competitive among existing PEFT methods, while slightly behind LoRI-D due to its more aggressive sparsity.

---

### Author Response · Authors · 2025-06-02
**Additional Baseline Results**

Dear Reviewers,

For your convenience, we have included the results of the added baselines in this response.

Table 1: Performance comparison of different adaptation methods on eight NLU benchmarks using Llama-3 with $r=32$. **Bold** indicates the best-performing method, and *italic* indicates the second-best.

| Method      | # Params (%)  |  BoolQ   |   PIQA   |   SIQA   |  ARC-c   |  ARC-e   |   OBQA   |  HellaS  |  WinoG   |   Avg.   |
| ----------- | ------------- | :------: | :------: | :------: | :------: | :------: | :------: | :------: | :------: | :------: |
| **FFT**     | 8.03G (100%)  |   73.8   |   86.8   |   77.6   |   76.7   |   87.6   |   84.1   |   93.2   |   85.1   |   83.1   |
| **LoRA**    | 84M (1.03%)   |  *76.3*  | **89.8** |   82.7   |   83.4   |   91.7   |   88.4   |   95.8   | **88.7** |  *87.1*  |
| **VeRA**    | 1.38M (0.02%) |   64.4   |   81.8   |   62.6   |   67.3   |   85.7   |   60.9   |   78.5   |   56.9   |   69.8   |
| **IA3**     | 1.70M (0.02%) |   68.6   |   84.8   |   74.5   |   77.6   |   89.4   |   75.7   |   90.6   |   75.0   |   79.5   |
| **LoRA-FA** | 44M (0.54%)   |   74.0   |  *89.6*  | **83.3** |  *83.8*  |  *93.4*  | **88.6** | **96.1** |   87.4   |   87.0   |
| **AdaLoRA** | 84M (1.03%)   |   75.6   |   89.2   |   82.4   |   83.1   |   91.0   |   87.8   |   94.4   |   87.6   |   86.4   |
| **rsLoRA**  | 84M (1.03%)   |   72.8   |   84.8   |   78.8   |   76.0   |   87.0   |   85.0   |   91.0   |   82.8   |   82.3   |
| **PiSSA**   | 84M (1.03%)   |   68.1   |   84.4   |   78.2   |   75.1   |   85.1   |   82.8   |   89.3   |   82.8   |   80.7   |
| **LoRA+**   | 84M (1.03%)   |   67.0   |   80.3   |   78.5   |   70.1   |   82.3   |   81.5   |   88.9   |   79.7   |   78.5   |
| **DoRA**    | 85M (1.05%)   |   75.9   | **89.8** |   82.7   |   83.5   |   93.2   |   87.9   |   95.3   |  *88.2*  |  *87.1*  |
| **LoRI-D**  | 44M (0.54%)   | **76.4** |   89.0   |   82.7   | **84.2** | **93.6** |  *88.5*  |  *95.9*  |   87.9   | **87.3** |
| **LoRI-S**  | 4.4M (0.05%)  |   75.2   |   89.2   |  *82.8*  |  *83.8*  |   92.6   |   88.4   |   95.2   |   87.5   |   86.8   |

Table 2: Performance comparison of different adaptation methods on GSM8K (math), HumanEval (code), and HEx-PHI (safety) benchmarks using Llama-3 with $r = 32$. **Bold** indicates the best-performing method, and *italic* indicates the second-best.

| Method      | # Params (%)  | GSM8K Accracy | HumanEval Pass@1 | HumanEval Pass@5 | HumanEval Pass@10 | HEx-PHI Score |
| ----------- | ------------- | :-----------: | :--------------: | :--------------: | :---------------: | :-----------: |
| **FFT**     | 8.03G (100%)  |     58.8      |       30.5       |       39.3       |       41.7        |     94.8      |
| **LoRA**    | 84M (1.03%)   |     64.4      |       34.7       |       46.4       |       50.8        |     91.6      |
| **VeRA**    | 1.38M (0.02%) |     30.6      |       32.4       |       45.1       |       50.9        |     74.7      |
| **IA3**     | 1.70M (0.02%) |     48.0      |       32.7       |       45.6       |       51.5        |     85.4      |
| **LoRA-FA** | 44M (0.54%)   |    *64.8*     |      *42.9*      |      *57.5*      |     **64.2**      |     94.1      |
| **AdaLoRA** | 84M (1.03%)   |     63.3      |       33.5       |       45.0       |       49.4        |     91.9      |
| **rsLoRA**  | 84M (1.03%)   |     61.3      |       28.4       |       35.5       |       38.3        |    *98.1*     |
| **PiSSA**   | 84M (1.03%)   |     61.3      |       32.0       |       40.3       |       43.3        |     97.8      |
| **LoRA+**   | 84M (1.03%)   |     61.7      |       33.0       |       42.7       |       46.0        |   **98.8**    |
| **DoRA**    | 85M (1.05%)   |   **65.4**    |       33.1       |       44.0       |       48.6        |     93.6      |
| **LoRI-D**  | 44M (0.54%)   |     63.2      |     **43.2**     |     **57.6**     |      *63.2*       |     92.8      |
| **LoRI-S**  | 4.4M (0.05%)  |     62.7      |       41.3       |       54.4       |       59.6        |     93.8      |


LoRI-D achieves a **50%** reduction in trainable parameters, while LoRI-S achieves a **95%** reduction, compared to LoRA. Despite tuning fewer parameters, LoRI-D and LoRI-S deliver comparable – and in many cases superior – performance across tasks in NLU, math, code, and safety domains.
The performance advantage is especially pronounced in NLU and code generation. These results highlight two key insights of our approach: (1) LoRI-D demonstrates that effective adaptation can be achieved without updating the A matrix, and (2) LoRI-S reveals substantial redundancy in the B matrix.

---

### Decision · Program_Chairs · 2025-07-08

**Decision:**

Accept

**Comment:**

The reviewers mostly believe the paper is worthy of publication at the conference. There were some additional experiments by the authors which strengthened their work and the response is satisfying.

[Automatically added comment] At least one review was discounted during the decision process due to quality]